# MINGLE: Mixture of Null-Space Gated Low-Rank Experts for Test-Time Continual Model Merging

**Zihuan Qiu**[1] **Yi Xu**[2] **Chiyuan He**[1] **Fanman Meng**[1*]
**Linfeng Xu**[1] **Qingbo Wu**[1] **Hongliang Li**[1]
[1]University of Electronic Science and Technology of China, Chengdu, China
[2]Dalian University of Technology, Dalian, China
{zihuanqiu@std., cyhe@std., fmmeng@, lfxu@, qbwu@, hlli@}uestc.edu.cn, yxu@dlut.edu.cn

## Abstract

Continual model merging integrates independently fine-tuned models sequentially without access to the original training data, offering a scalable and efficient solution for continual learning. However, existing methods face two critical challenges: parameter interference among tasks, which leads to catastrophic forgetting, and limited adaptability to evolving test distributions. To address these issues, we introduce the task of Test-Time Continual Model Merging (TTCMM), which leverages a small set of unlabeled test samples during inference to alleviate parameter conflicts and handle distribution shifts. We propose MINGLE, a novel framework for TTCMM. MINGLE employs a mixture-of-experts architecture with parameter-efficient, low-rank experts, which enhances adaptability to evolving test distributions while dynamically merging models to mitigate conflicts. To further reduce forgetting, we propose Null-Space Constrained Gating, which restricts gating updates to subspaces orthogonal to prior task representations, thereby suppressing activations on old tasks and preserving past knowledge. We further introduce an Adaptive Relaxation Strategy that adjusts constraint strength dynamically based on interference signals observed during test-time adaptation, striking a balance between stability and adaptability. Extensive experiments on standard continual merging benchmarks demonstrate that MINGLE achieves robust generalization, significantly reduces forgetting, and consistently surpasses previous state-of-the-art methods by 7–9% on average across diverse task orders. Our code is available at:
`https://github.com/zihuanqiu/MINGLE`

## 1 Introduction

Continual learning aims to incrementally adapt machine learning models to new tasks without forgetting previously learned knowledge, addressing the critical challenge of catastrophic forgetting [43]. However, conventional continual learning approaches typically require continuous access to original training data, raising significant concerns about privacy and substantial computational overhead due to retraining efforts, thus limiting their applicability in dynamic, data-sensitive environments.

To address these limitations, recent works have explored an alternative paradigm known as continual model merging (CMM), which sequentially integrates independently fine-tuned models directly in parameter space, without revisiting any training data [38, 51, 70]. CMM typically operates under a "merge-to-transfer" paradigm: given a pretrained model $\theta_0$ and independently fine-tuned models $\{\theta_t\}_{t=1}^T$, a unified model is constructed sequentially by combining task-specific weight updates $\Delta\theta_t = \theta_t - \theta_0$ via weighted averaging or projection-based strategies [24, 81, 69].

---

*Corresponding author

Despite its advantages in scalability, data privacy, and distributed training capabilities [44, 14, 57], existing CMM methods still encounter critical issues, notably severe parameter interference between tasks and limited adaptability to evolving test distributions. This parameter interference arises because, as fine-tuned models are incrementally merged, overlapping or conflicting parameter updates accumulate, resulting in severe forgetting of previously learned tasks. To mitigate this interference, recent methods introduce structural constraints such as orthogonal projection [70, 78], model linearization [38, 69], and pruning-based sparsification [86, 90]. However, their effectiveness diminishes as task count grows and interference becomes increasingly entangled. Moreover, models merged across tasks often fail to generalize effectively, particularly when facing unseen or shifting task conditions. These flaws result in severe forgetting of earlier tasks and substantial performance gaps compared to the upper bound achieved by individually fine-tuned models. As shown in Fig. 1, TA [24] suffers from large performance gaps and strong forgetting, reflected by low accuracy and negative backward transfer. OPCM [70] improves over TA via orthogonalized merging but still shows notable degradation.

To overcome these limitations, we propose a novel continual merging paradigm—**Test-Time Continual Model Merging (TTCMM)**—which explicitly introduces the concept of test-time adaptation (TTA) [75, 63] into model merging. Unlike prior TTA-based multi-task merging methods [87, 68, 88], which assume simultaneous availability of models and test data from all tasks, TTCMM utilizes only a small set of unlabeled samples from the current task, making it uniquely suited for realistic continual scenarios where revisiting historical data is often infeasible.

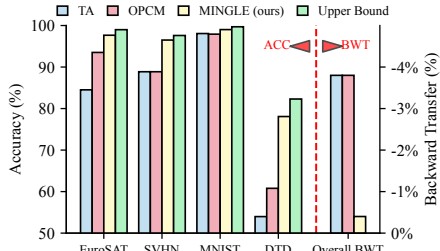

Figure 1: After 8-task continual merging: accuracy on first four tasks and overall BWT.

In this paper, we propose MINGLE (**MI**xture of **N**ull-Space **G**ated **L**ow-Rank **E**xperts), a method designed to continually merge independently fine-tuned models at test-time while preserving prior knowledge. MINGLE employs a mixture-of-experts architecture [27, 45] composed of lightweight LoRA-based [21] experts, enabling efficient and flexible test-time adaptation. To robustly prevent interference from previously learned tasks, we introduce a novel **Null-Space Constrained Gating** mechanism, restricting gating updates to task-orthogonal subspaces. Additionally, we propose an **Adaptive Relaxation Strategy** to dynamically modulate constraint strength based on test-time interference feedback during adaptation.

Extensive experiments on standard continual learning benchmarks show that MINGLE consistently outperforms previous state-of-the-art approaches by 7–9% on average, achieving robust generalization and strong resistance to catastrophic forgetting across diverse continual learning scenarios. Remarkably, these improvements are achieved entirely without any access to original training data, demonstrating the effectiveness of our TTCMM paradigm and the power of test-time adaptation in continual learning.

Our contributions are summarized as follows:

- We formalize test-time continual model merging (TTCMM), a novel task that leverages unlabeled test samples to merge independently fine-tuned models.

- We propose MINGLE, a TTCMM framework with Adaptive Null-Space Constrained Gating to effectively balance stability and plasticity.

- Extensive experiments show that MINGLE achieves state-of-the-art performance, consistently outperforming prior methods in accuracy, robustness and resistance to forgetting.

## 2 Related Work

**Continual Learning.** Continual learning (CL) seeks to mitigate catastrophic forgetting [43], where learning new tasks overwrites prior knowledge. Regularization-based methods constrain updates with importance weights [31, 92, 2, 30, 82], while distillation aligns outputs to preserve knowledge [20, 12, 60, 52]. Replay methods store exemplars or generate surrogates with prompts, prototypes, or generators [55, 39, 79, 61, 53], and dynamic architectures expand capacity via growth or ensembling [35, 98, 42]. Recent work leverages lightweight adapters or prompts in pre-trained models for efficient transfer [89, 23, 80]. Model merging offers an alternative route. Some methods remain close to

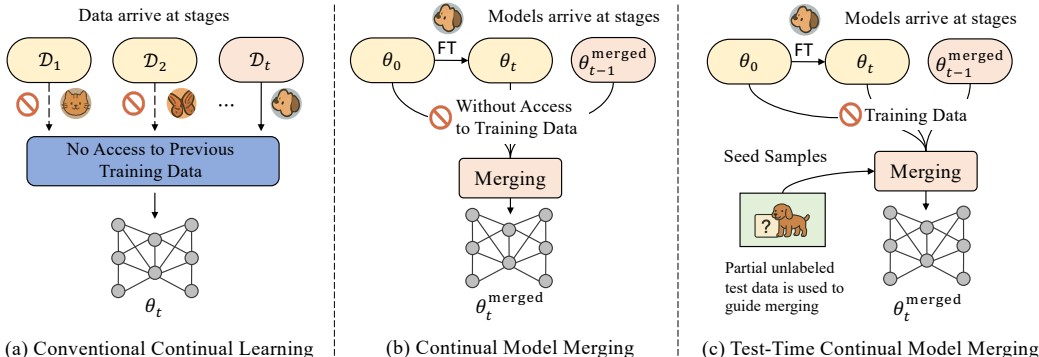

Figure 2: Comparison of three continual learning paradigms. (a) Conventional Continual Learning trains models sequentially with data arriving in stages, without access to previous task data. (b) Continual Model Merging continually fuses independently trained models, without access to any training data. (c) Test-Time Continual Model Merging improves merging by leveraging a few unlabeled test samples from the current task.

conventional CL by sequentially fine-tuning and merging models to reduce forgetting [41, 42, 16], typically requiring training data. In contrast, continual model merging [28, 38, 5, 51, 70] merges *independently fine-tuned models* without revisiting training data, enabling greater scalability and privacy.

**Model Merging.** Early work merged models via direct parameter averaging [72, 58], later refined by linear mode connectivity [13, 1]. Wortsman *et al.* [81] showed that weight averaging can also enhance robustness and out-of-distribution generalization. Task Arithmetic (TA) [24] views models as task vectors to be summed, but relies on weight disentanglement [49], often violated under standard fine-tuning, motivating structured training [28, 65]. Beyond averaging, interference-aware methods reweight or sparsify parameters [86, 90], or fuse models via distillation and clustering [88, 76]. LoRA-based tuning [21] introduces additional entanglement challenges, spurring gradient-free or retrieval-based strategies [22, 94, 95]. More recently, dynamic merging [68, 40] adapts parameters conditioned on inputs, achieving higher flexibility and performance, but remains limited to multi-task fusion and unexplored in continual settings.

**Test-Time Adaptation.** TTA adapts models at inference to mitigate distribution shift. Early approaches used self-supervised objectives [66], entropy minimization [75], or regularized updates [63]. Online TTA adapts continuously [25], while batch-wise variants ignore temporal structure [17]. To enhance stability, later work introduced confidence filtering [48], EMA [11], partial updates [91], test-time augmentation [93], and adaptive BatchNorm [56]. For vision–language models, TTA often employs prompts or adapters [59, 15, 36]. We draw on TTA to guide merging, aligning fused models with evolving test distributions.

**Relation to Prior Work.** Most related to our work are MoE-Adapter [89] and WEMOE [68], both built on MoE architectures [27, 29]. MoE-Adapter follows conventional continual learning, embedding expert modules that are jointly trained across tasks. In contrast, we adopt a model-merging paradigm, where experts are extracted from independently fine-tuned models and inserted without further training. WEMOE incorporates test-time adaptation but targets multitask learning, assuming simultaneous access to all models and data. By contrast, MINGLE is tailored for the more challenging continual merging setting.

## 3 MINGLE: Mixture of Null-Space Gated Low-Rank Experts

### 3.1 Preliminaries

**Problem Setting.** We study continual learning in a model merging setting, where a sequence of task-specific models $\{\theta_1, \ldots, \theta_T\}$ are independently fine-tuned from a shared pre-trained model $\theta_0$, each using a dataset $\mathcal{D}_i = \{(x_j^{(i)}, y_j^{(i)})\}$ with label space $\mathcal{C}_i \subset \mathcal{Y}$. The goal is to construct a unified model $\theta_T^{\text{merged}}$ that generalizes across the combined label space $\mathcal{C}_{1:T} = \bigcup_{i=1}^{T} \mathcal{C}_i$.

Unlike conventional continual learning, we assume *no access* to training data during merging. All adaptation happens directly in parameter space. This paradigm is relevant in scenarios where only final fine-tuned models are retained, while original training data is discarded due to privacy, storage, or accessibility constraints.

To contextualize this, we compare with two related paradigms in Fig. 2:

- **Conventional Continual Learning.** A single model $\theta$ is sequentially updated on $\mathcal{D}_1, \ldots, \mathcal{D}_T$, discarding previous data. It requires direct training data access and extensive retraining.

- **Continual Model Merging.** A sequence of models are merged incrementally in parameter space without access to training data and earlier models: $\theta_t^{\text{merged}} = \text{Merge}(\theta_{t-1}^{\text{merged}}, \theta_t)$.

- **Test-Time Continual Model Merging.** An extension of the above where a small unlabeled subset $\mathcal{D}_t^{\text{seed}} \subset \mathcal{D}_t^{\text{test}}$ (*e.g.*, 5 samples per class) is available at each stage to provide lightweight task-specific guidance. We refer to $\mathcal{D}_t^{\text{seed}}$ as the *seed samples* of task $t$.

**Existing Continual Merging Strategies.** Let $\theta_0$ denote the parameters of a pre-trained model. The corresponding task vector is defined as $\Delta \theta_t = \theta_t - \theta_0$.

- **Continual Task Arithmetic (C. TA).** A simple additive merge [24]: $\theta_t^{\text{merged}} = \theta_{t-1}^{\text{merged}} + \lambda \Delta \theta_t$, where $\lambda$ is a scalar. While training-free, it is sensitive to $\lambda$ and prone to task interference.

- **Orthogonal Projection-based Continual Merging (OPCM).** Tang et al. [70] propose projecting each $\Delta \theta_t$ onto the orthogonal complement of previous directions: $\theta_t^{\text{merged}} = \theta_0 + \frac{1}{\lambda_t} \left[ \lambda_{t-1} \Delta \theta_{t-1}^{\text{merged}} + \mathcal{P}^{(t-1)}(\Delta \theta_t) \right]$, where $\mathcal{P}^{(t-1)}$ retains components orthogonal to previous updates. This reduces interference but ignores adaptation to task distributions.

To address these issues, we present MINGLE, which leverages $\mathcal{D}_t^{\text{seed}}$ to modulate the integration of $\theta_t$ at test-time, enhancing alignment to test distribution and mitigating task interference.

## 3.2 Motivation and Theoretical Analysis

Most existing continual model merging methods combine fine-tuned models via static averaging, where each expert is assigned fixed coefficients, thereby enforcing the *same* mixing rule across the whole input space. Consequently, it cannot specialize to regions where one expert is clearly superior. In contrast, a Mixture-of-Experts (MoE) equips every input with a *data-dependent* gate $g(x) = (g_1(x), \ldots, g_T(x))$ that selects or re-weights experts on-the-fly. We give a formal comparison between static averaging and dynamic MoE under a noisy-routing scenario. [2]

**Theorem 1** (Dynamic MoE versus Static Averaging). *Let $\{(D_t, f_t)\}_{t=1}^T$ be $T$ independent tasks with priors $P(t)$ and per-task risks $R_t(i)$. For any static mixture $h_{\text{static}}(x) = \sum_{i=1}^T \alpha_i f_i(x)$ and any hard-routed MoE $h_{\text{MoE}}(x) = f_{i^\star(x)}(x)$ with task-specific routing errors $\varepsilon_t$:*

$$R(h_{\text{MoE}}) = R_{\text{ideal}} + \sum_{t=1}^T P(t)\, \varepsilon_t \big(R_{wrong,t} - R_t(t)\big), \tag{1}$$

*where $R_{\text{ideal}} = \sum_t P(t) R_t(t)$ and $R_{wrong,t} = \frac{1}{T-1} \sum_{i \neq t} R_t(i)$. Moreover,*

1. *(Perfect routing) If $\varepsilon_t = 0$ for all $t$, then $\inf_g R(h_{\text{MoE}}) < \inf_{\boldsymbol{\alpha}} R(h_{\text{static}})$ whenever at least two tasks disagree on their best expert.*

2. *(Noisy routing) If $\sum_t P(t) \varepsilon_t \big(R_{wrong,t} - R_t(t)\big) < R_{\text{static}}^* - R_{\text{ideal}}$, where $R_{static}^* = \inf_{\boldsymbol{\alpha}} R(h_{static})$, then the MoE still attains lower risk than any static mixture.*

The theory above motivates a design that (i) keeps experts specialized and (ii) prevents interference between tasks. Our MINGLE framework achieves both goals by combining

- **Low-rank experts** $f_t$ that capture task-specific variations with minimal parameters, and

- **Null-space constrained gating** that projects gradient updates away from subspaces spanned by previously activated features, keeping $\varepsilon_t$ small without harming earlier experts.

---

[2]Symbols and proofs are deferred to App. A

### 3.3 Low-Rank Expert Mixture for Continual Model Merging

We adopt MoE framework for continual model merging, in which each task $i$ is equipped with a low-rank expert $f_i$ and an associated input-dependent gating function $g_i$. These components are injected into the linear layers of the backbone (*e.g.*, CLIP visual encoder). The gate $g_i$ modulates expert activation based on the input features, allowing for fine-grained, localized task specialization.

**Mixture of Low-Rank Expert.** When a new task $t$ arrives, a dedicated expert $f_t$ and its gate $g_t$ are appended to the model. The output of a given $l$-th layer[3] can be formulated as follows:

$$\theta_t^{\text{merged},(l)}(X) = \theta_{t-1}^{\text{merged},(l)}(X) + g_t^{(l)}(X) \cdot f_t^{(l)}(X) = \theta_0^{(l)}(X) + \sum_{i=1}^{t} g_i^{(l)}(X) \cdot f_i^{(l)}(X). \quad (2)$$

where only the gate $g_t$ is adaptable during testing, while all experts $\{f_i\}_{i=1}^{t}$ and old gates $\{g_i\}_{i=1}^{t-1}$ remain frozen to preserve prior knowledge. To construct expert $f_t$, we first project the task vector $\Delta\theta_t$ onto the orthogonal complement of previously learned directions, following OPCM [70]:

$$\mathcal{P}^{(t-1)}(\Delta\theta_t) = \sum_{p,q=a,p\neq q}^{m,n} \langle \Delta\theta_t, u_p^{(t-1)} v_q^{(t-1)\top} \rangle_F u_p^{(t-1)} v_q^{(t-1)\top}, \quad (3)$$

where $u_p^{(t-1)}$ and $v_q^{(t-1)}$ are the $p$-th and $q$-th singular vectors from the singular value decomposition (SVD) of previous experts $\sum_{i=1}^{t-1} f_i(X)$, and $\alpha$ denotes the effective rank of previous experts. This projection removes previously learned directions to mitigate interference. We then apply a rank-$r$ truncated SVD for $\mathcal{P}^{(t-1)}(\Delta\theta_t)$ to construct a low-rank expert [21]:

$$f_t = BA = (\tilde{U}\tilde{\Sigma})\tilde{V}^\top, \quad (4)$$

where $\tilde{U} \in \mathbb{R}^{d_1 \times r}$, $\tilde{\Sigma} \in \mathbb{R}^{r \times r}$, and $\tilde{V} \in \mathbb{R}^{d_2 \times r}$, retaining the top $r$ singular components. The resulting expert captures the principal directions while significantly reducing parameter overhead.

Each gating function is implemented as a linear projection:

$$g_t(X) = W_t^{(g)\top} X + b_t^{(g)}, \quad (5)$$

where $W_t^{(g)} \in \mathbb{R}^{d \times 1}$ and $b_t^{(g)} \in \mathbb{R}$ are *learnable* parameters. The gating function is adapted at test time using a small number of unlabeled test data.

**Test-Time Adaptation.** To encourage the merged model to retain task-specific behavior, we minimize the Kullback–Leibler divergence between its prediction and that of the corresponding individual fine-tuned model $\theta_t$. We define the adaptation objective as:

$$L_t = \mathbb{E}_{x \sim \mathcal{D}_t^{\text{seed}}} \left[ \text{KL} \left( p(x; \theta_t^{\text{merged}}) \, \| \, p(x; \theta_t) \right) \right]. \quad (6)$$

where $p(x; \theta)$ denotes the predictive distribution

### 3.4 Adaptive Null-Space Constrained Gating for Interference Mitigation

When merging models continually, the primary challenge of gating is to integrate new experts without disturbing prior task predictions. Consider two experts $f_1, f_2$ and their corresponding gates $g_1, g_2$. When evaluating on the first task domain $X_1$, the interference from $g_2$ can be quantified as:

$$\xi(g_2) = \|g_1(X_1) \cdot f_1(X_1) + g_2(X_1) \cdot f_2(X_1) - f_1(X_1)\|^2. \quad (7)$$

This measures the deviation introduced by $g_2$ on the domain where $f_1$ originally dominates. A desirable gating function should suppress $g_2(X_1)$, ensuring predictions on $X_1$ remain unaffected. However, as $X_1$ becomes inaccessible after adaptation, this error becomes unobservable and cannot be minimized directly, resulting in prediction drift and catastrophic forgetting.

**Hard Null-Space Projection.** After completing task $t$, we cache the $l$-th layer inputs in the seed buffer $\mathcal{D}_t^{\text{seed}}$ and estimate their covariance $\text{Cov}_t^{(l)} \in \mathbb{R}^{d \times d}$. Applying truncated SVD yields the

---

[3]The layer index $l$ is omitted hereafter whenever it does not cause ambiguity.

top-$k$ dominant subspaces $\tilde{U}_t^{(l)} \in \mathbb{R}^{d \times k}$. We then concatenate these with the subspaces from all previous tasks and orthonormalize: $U_t^{(l)} = \mathrm{orthonorm}\big[U_{t-1}^{(l)} | \tilde{U}_t^{(l)}\big] \in \mathbb{R}^{d \times tk}$. The hard projector is $P_t = I - U_t^{(l)} U_t^{(l)\top} \in \mathbb{R}^{d \times d}$. To suppress interference from tasks $\leq t-1$, the gating update for task $t$ is $W_t^{(g,l)} \leftarrow W_t^{(g,l)} - \eta \nabla L_t^{(l)} P_{t-1}^{(l)}$. However, this projection may also discard gradient components that are informative for task $t$ whenever its feature support overlaps with $\mathrm{span}(U_{t-1})$.

**Adaptive Null-Space Relaxation.** To restore plasticity, we replace the *all-one* eigenvalues of $P_{t-1}$ with *soft* coefficients learned online.

*(i) Interference statistics.* For each column $u_p^{(l)}$ of $U_{t-1}^{(l)}$ we measure instantaneous alignment:

$$r_p^{(l)} = \|(\nabla L_t)^\top u_p^{(l)}\|_2 \, / \, \|\nabla L_t\|_2. \quad (8)$$

We maintain per-direction interference scores $S^{(m,l)} \in \mathbb{R}^k$ (initialized to 0) by applying an exponential moving average at each iteration $m$:

$$S^{(m,l)} = \beta \, S^{(m-1,l)} + (1-\beta) \, r^{(l)}, \quad (9)$$

which suppresses stochastic gradient noise while preserving the dominant interference directions.

*(ii) Adaptive shrinkage.* Each direction is attenuated by $\lambda^{(m,l)} = \exp(-\gamma S^{(m,l)})$ $(\gamma > 0, \; \lambda^{(m,l)} \in (0,1])$. Let $\Lambda_{t-1}^{(m,l)} = \mathrm{diag}(\lambda_1^{(m,l)}, \ldots, \lambda_{(t-1)\cdot k}^{(m,l)})$. The *relaxed projector* becomes:

$$\widetilde{P}_{t-1}^{(m,l)} = U_{t-1}^{(m,l)} \Lambda_{t-1}^{(m,l)} U_{t-1}^{(m,l)\top}, \quad (10)$$

interpolating smoothly between no protection $(\Lambda = 0)$ and the hard null projector $(\Lambda = I)$.

*(iii) Update rule.* We finally update:

$$W_t^{(g,l)} \leftarrow W_t^{(g,l)} - \eta \nabla L_t^{(l)} \widetilde{P}_{t-1}^{(l)}. \quad (11)$$

---

**Algorithm 1** MINGLE Procedure.

**Input:** pre-trained model $\theta_0$ and fine-tuned models $\{\theta_t\}_{t=1}^T$; seed data $\{\mathcal{D}_t^{\mathrm{seed}}\}_{t=1}^T$; hyperparameters $k, \beta, \gamma$; learning rate $\eta$

**Output:** Merged model $\theta_T^{\mathrm{merged}}$

**Init:** $\theta_0^{\mathrm{merged}} \leftarrow \theta_0$; $U_0^{(l)} \leftarrow \emptyset$

**for** task $t = 1$ **to** $T$ **do**
  ▷ create low-rank experts (Eqs. 3 and 4)
  $f_t = \mathrm{SVD_{TRUNC.}} \big( \underbrace{\mathcal{P}^{(t-1)}(\theta_t - \theta_0)}_{\text{use } \theta_1 - \theta_0 \text{ when } t=1} \big) = B_t A_t$
  ▷ add expert & initialize gate (Eq. 5)
  $\theta_t^{\mathrm{merged}} = \theta_{t-1}^{\mathrm{merged}} + g_t \cdot f_t$, $\{W^{(g)}, b^{(g)}, S^{(0)}\} \leftarrow 0$
  **for** $m = 1$ **to** *total iterations* **do**
    $X \leftarrow \mathrm{batch}(\mathcal{D}_t^{\mathrm{seed}})$
    $\{\nabla_{W_t^{(g)}} L, \nabla_{b_t^{(g)}} L\} \leftarrow \nabla L_t(X, \theta_t^{\mathrm{merged}}, \theta_t)$
    **if** $t > 1$ **then**
      ▷ project gradient onto null-space
        (Eqs. 8-11)
      $S^{(m)} = \beta S^{(m-1)} + (1-\beta)r$
      $\nabla_{W_t^{(g)}} L_t \leftarrow \nabla_{W_t^{(g)}} L_t \widetilde{P}_{t-1}$
    **end**
    $W_t^{(g)} \leftarrow W_t^{(g)} - \eta \nabla_{W_t^{(g)}} L_t$
    $b_t^{(g)} \leftarrow b_t^{(g)} - \eta \nabla_{b_t^{(g)}} L_t$
  **end**
  ▷ update dominant subspaces
  $U_t \leftarrow \mathrm{orthonorm}[U_{t-1} | \tilde{U}_t]$
**end**

---

Relaxing the projector inevitably allows more residual interference than the hard null-space variant, yet empirically the increase is minor and is offset by markedly higher plasticity (Tab. 5), indicating a favorable *stability–plasticity balance*. The overall procedure is outlined in Algo. 1.

## 4 Experiments

We describe the experimental setup in Sec. 4.1, followed by the main results in Sec. 4.2 and further analysis and ablations in Sec. 4.3. Due to page limitations, detailed results are provided in the Appendix.

### 4.1 Experimental Setup

**Datasets and Models.** Following [70], we evaluate on image-classification tasks with CLIP-ViT backbones [54]. We consider 8, 14, and 20-task groups using ViT-B/32, ViT-B/16, and ViT-L/14 models, each fine-tuned on up to 20 downstream tasks, with checkpoints from FusionBench [67]. To assess order sensitivity, we repeat experiments over 10 random seeds (42–51). For comparison with conventional CL, we use the Multi-domain Task-Incremental Learning (MTIL) benchmark [97] with eleven vision tasks. Beyond vision, we evaluate on eight GLUE language tasks [74] with a Flan-T5-base backbone [6].

Table 1: Comparative results of continual merging methods, reporting average accuracy (ACC) and backward transfer (BWT) over ten task orders (mean±std). DM and DA denote method assumptions: dynamic merging or test data access. Best results are in bold; second-best are underlined. MINGLE* denotes a lightweight variant.

| Method | Assump. DM / DA | ViT-B/32 | | | ViT-B/16 | | | ViT-L/14 | | |
|---|---|---|---|---|---|---|---|---|---|---|
| | | 8 tasks | 14 tasks | 20 tasks | 8 tasks | 14 tasks | 20 tasks | 8 tasks | 14 tasks | 20 tasks |
| PRE-TRAINED | – / – | 48.1 | 56.9 | 55.6 | 55.4 | 62.0 | 59.8 | 64.9 | 69.1 | 65.6 |
| FINE-TUNED | – / – | 90.4 | 89.3 | 89.8 | 92.4 | 91.3 | 91.6 | 94.3 | 93.4 | 93.5 |
| C. FINE-TUNED | – / – | 79.8 | 67.4 | 62.6 | 82.9 | 72.2 | 68.2 | 90.0 | 70.9 | 77.7 |
| **ACC (%) ↑** | | | | | | | | | | |
| AVERAGE (SWA) [26] | ✗ / ✗ | 66.3 ±0.0 | 65.4 ±0.0 | 61.1 ±0.0 | 72.3 ±0.0 | 69.7 ±0.0 | 64.8 ±0.0 | 80.0 ±0.0 | 77.5 ±0.0 | 71.1 ±0.0 |
| C. TASK ARITHMETIC [24] | ✗ / ✗ | 67.5 ±0.0 | 66.5 ±0.0 | 60.0 ±0.0 | 77.1 ±0.0 | 70.9 ±0.6 | 64.2 ±0.0 | 82.1 ±0.0 | 77.9 ±0.0 | 70.3 ±0.0 |
| C. TIES-MERGING [86] | ✗ / ✗ | 49.0 ±10.2 | 66.2 ±0.6 | 59.9 ±0.7 | 66.8 ±3.7 | 70.5 ±0.8 | 63.0 ±1.6 | 64.3 ±7.0 | 78.0 ±0.6 | 68.3 ±0.9 |
| MAGMAX-IND [41] | ✗ / ✗ | 70.7 ±0.0 | 67.0 ±0.0 | 61.2 ±0.0 | 76.7 ±1.8 | 67.0 ±0.0 | 62.5 ±0.0 | 83.4 ±0.0 | 71.2 ±0.0 | 71.2 ±0.0 |
| CONSENSUS TA [77] | ✗ / ✗ | 67.1 ±0.4 | 64.1 ±0.8 | 45.8 ±1.5 | 72.8 ±0.5 | 69.0 ±0.0 | 49.9 ±1.9 | 80.4 ±0.5 | 75.0 ±1.0 | 51.3 ±2.4 |
| OPCM [70] | ✗ / ✗ | 75.5 ±0.5 | 71.9 ±0.3 | 65.7 ±0.2 | 81.8 ±0.3 | 77.1 ±0.5 | 70.3 ±0.2 | 87.0 ±0.4 | 83.5 ±0.2 | 76.0 ±0.2 |
| C. LW ADAMERGING [87] | ✗ / ✓ | 53.4 ±3.2 | 59.8 ±1.6 | 59.7 ±7.4 | 59.9 ±2.3 | 64.3 ±1.2 | 61.5 ±1.1 | 68.8 ±2.9 | 73.1 ±5.7 | 66.9 ±1.1 |
| C. LORA-WEMOE [68] | ✓ / ✓ | 68.8 ±7.8 | 63.8 ±3.4 | 49.6 ±15.4 | 72.6 ±3.7 | 67.9 ±2.9 | 55.0 ±7.0 | 75.6 ±7.8 | 74.0 ±5.0 | 56.9 ±19.8 |
| MINGLE (Ours) | ✓ / ✓ | **85.8** ±0.8 | 81.6 ±1.4 | **77.1** ±2.0 | **88.3** ±0.6 | **84.9** ±0.8 | **81.9** ±0.9 | **91.8** ±0.2 | 88.8 ±0.7 | **85.5** ±1.3 |
| MINGLE* (Ours) | ✓ / ✓ | 85.0 ±0.5 | **81.7** ±1.0 | 77.1 ±1.3 | 87.0 ±0.6 | 84.7 ±1.0 | 81.6 ±1.3 | 91.4 ±0.3 | **89.2** ±0.1 | 83.6 ±0.6 |
| **BWT (%) ↑** | | | | | | | | | | |
| AVERAGE (SWA) [26] | ✗ / ✗ | -11.5 ±2.2 | -8.0 ±1.3 | -7.1 ±2.1 | -9.7 ±1.5 | -7.1 ±1.4 | -7.3 ±1.7 | -7.3 ±1.4 | -5.8 ±1.0 | -6.4 ±1.5 |
| C. TASK ARITHMETIC [24] | ✗ / ✗ | -9.6 ±1.5 | -1.3 ±1.6 | -3.4 ±1.0 | -4.2 ±1.0 | -1.3 ±0.4 | -3.6 ±0.4 | -7.1 ±0.8 | -1.8 ±0.3 | -3.3 ±0.3 |
| C. TIES-MERGING [86] | ✗ / ✗ | -15.3 ±8.0 | **1.9** ±0.6 | -1.5 ±0.7 | -5.5 ±0.4 | **1.4** ±0.7 | -1.5 ±1.2 | -13.0 ±5.7 | -1.1 ±0.4 | -2.9 ±1.0 |
| MAGMAX-IND [41] | ✗ / ✗ | -8.3 ±1.3 | -7.4 ±1.4 | -7.2 ±1.6 | -6.1 ±1.3 | -7.4 ±2.0 | -8.0 ±2.2 | -5.0 ±0.8 | -6.0 ±2.1 | -6.5 ±2.1 |
| CONSENSUS TA [77] | ✗ / ✗ | **3.8** ±0.9 | -1.3 ±0.9 | -11.8 ±1.9 | **3.5** ±0.6 | -1.1 ±0.8 | -11.6 ±1.3 | **2.4** ±0.6 | -2.5 ±0.8 | -16.5 ±1.5 |
| OPCM [70] | ✗ / ✗ | -6.3 ±1.1 | -6.0 ±1.0 | -7.8 ±1.5 | -4.8 ±0.7 | -5.1 ±1.4 | -6.3 ±2.2 | -2.6 ±1.0 | -4.3 ±0.7 | -6.5 ±1.8 |
| C. LW ADAMERGING [87] | ✗ / ✓ | -32.5 ±3.6 | -24.1 ±1.7 | -22.7 ±4.3 | -27.8 ±2.7 | -22.1 ±1.4 | -21.4 ±1.2 | -24.3 ±3.3 | -19.6 ±1.7 | -21.7 ±1.1 |
| C. LORA-WEMOE [68] | ✓ / ✓ | -20.4 ±9.0 | -20.2 ±3.9 | -24.5 ±10.0 | -18.0 ±6.2 | -18.8 ±3.4 | -25.8 ±7.9 | -17.8 ±5.9 | -16.8 ±5.3 | -27.9 ±17.2 |
| MINGLE (Ours) | ✓ / ✓ | -0.6 ±0.4 | -1.1 ±0.3 | -2.2 ±0.8 | -0.4 ±0.1 | -0.9 ±0.1 | -1.9 ±0.4 | -0.6 ±0.1 | -1.0 ±0.3 | -2.6 ±0.9 |
| MINGLE* (Ours) | ✓ / ✓ | -0.1 ±0.1 | -0.4 ±0.1 | **-1.3** ±0.6 | -0.1 ±0.1 | -0.3 ±0.1 | **-1.0** ±0.4 | -0.2 ±0.0 | **-0.4** ±0.2 | **-1.5** ±0.6 |

Table 2: Results of continual merging Flan-T5-base models on 8 tasks, ordered alphabetically.

| Method | DM / DA | CoLA | MNLI | MRPC | QNLI | QQP | RTE | SST2 | STSB | ACC ↑ | BWT ↑ |
|---|---|---|---|---|---|---|---|---|---|---|---|
| PRE-TRAINED | – / – | 69.1 | 56.5 | 76.2 | 88.4 | 82.1 | 80.1 | 91.2 | 62.2 | 75.7 | - |
| INDIVIDUAL | – / – | 75.0 | 83.4 | 87.5 | 91.5 | 85.4 | 85.9 | 93.6 | 88.7 | 86.4 | - |
| TASK ARITHMETIC | ✗ / ✗ | 69.1 | 58.1 | 77.9 | 88.9 | 83.1 | 79.1 | 90.7 | 74.0 | 77.6 | -4.6 |
| TIES-MERGING | ✗ / ✗ | 39.3 | 70.0 | 82.4 | 88.8 | 81.8 | 75.8 | 89.7 | 76.8 | 75.6 | -6.1 |
| OPCM | ✗ / ✗ | 69.9 | 72.9 | 78.7 | 90.3 | 83.8 | **83.0** | 92.2 | 73.7 | 80.6 | -2.5 |
| LW ADAMERGING | ✗ / ✓ | 69.1 | 58.1 | 77.9 | 88.9 | 83.1 | 79.1 | 90.7 | 74.2 | 77.6 | -4.7 |
| LORA-WEMOE | ✓ / ✓ | 71.5 | **80.6** | 78.2 | 90.3 | 82.7 | 80.5 | 91.3 | 76.2 | 81.4 | **0.1** |
| MINGLE (Ours) | ✓ / ✓ | **75.0** | 78.2 | **86.0** | **90.9** | **84.2** | 80.5 | **92.5** | 78.8 | **83.3** | **0.1** |

**Implementation Details.** We insert low-rank experts into the CLIP vision encoder. Two variants are used: a full setup modifying all attention and MLP layers, and a lightweight one on `attn.qkv` and `mlp.fc1`. All experiments share a single set of *global* hyper-parameters across models and task orders. Each expert has rank $r = 64$; the null-space constraint uses $k = 3$, $\gamma = 1$, and $\beta = 0.99$. Adaptation runs for 50 iterations with Adam (lr 1e-4, batch size 16). For vision tasks we use 5 unlabeled samples per class, and for NLP tasks 100 in total, all without access to prior-task data.

**Evaluation Metrics.** We evaluate using average accuracy (ACC) and backward transfer (BWT) [37]. ACC is the mean accuracy of the final merged model across all tasks: $\text{ACC} = \frac{1}{T} \sum_{i=1}^{T} a_i(\theta_t^{\text{merged}})$, where $a_i(\cdot)$ is accuracy on task $i$. BWT measures forgetting by comparing performance on earlier tasks before vs. after the final merge: $\text{BWT} = \frac{1}{T-1} \sum_{i=1}^{T-1} \left[ a_i(\theta_T^{\text{merged}}) - a_i(\theta_i^{\text{merged}}) \right]$.

## 4.2 Main Results

**Overall Performance** As shown in Tab. 1, MINGLE substantially outperforms previous continual merging methods on all CLIP backbones and task counts. It achieves the highest accuracy with backward forgetting kept near zero, demonstrating both strong forward learning and long-term stability. The lightweight variant performs on par with the full version, further underscoring the robustness of our approach. On NLP benchmarks (Tab. 2), MINGLE likewise attains the best overall accuracy and non-negative BWT, improving on multiple GLUE tasks while maintaining balanced performance across the suite. Together, these results across vision and language confirm that MINGLE

Table 3: Comparison of last accuracy (%) with conventional CL approaches on MTIL benchmark.

| Method | Aircraft | Caltech101 | CIFAR100 | DTD | EuroSAT | Flowers | Food101 | MNIST | Pets | Cars | SUN397 | Avg. |
|---|---|---|---|---|---|---|---|---|---|---|---|---|
| **Conventional CL** (*Sequential fine-tuned*) | | | | | | | | | | | | |
| WiSE-FT [81] | 27.2 | 90.8 | 68.0 | 68.9 | 86.9 | 74.0 | 87.6 | **99.6** | 92.6 | 77.8 | 81.3 | 77.7 |
| ZSCL [97] | 40.6 | 92.2 | 81.3 | 70.5 | 94.8 | 90.5 | 91.9 | 98.7 | 93.9 | 85.3 | 80.2 | 83.6 |
| MoE-Adapter [89] | 49.8 | 92.2 | 86.1 | 78.1 | 95.7 | 94.3 | 89.5 | 98.1 | 89.9 | 81.6 | 80.0 | 85.0 |
| DIKI [71] | 45.2 | 95.7 | 86.3 | 72.9 | **98.0** | 97.0 | 89.2 | 99.4 | **94.2** | 81.6 | 76.6 | 85.1 |
| AwoForget [96] | 42.4 | 92.7 | 83.2 | 73.2 | 97.0 | 91.8 | **92.2** | 99.1 | 93.9 | **87.4** | **82.6** | 85.0 |
| Dual-RAIL [85] | 52.5 | 96.8 | 83.3 | **80.1** | 96.4 | **99.0** | 89.9 | 98.8 | 93.5 | 85.5 | 79.2 | **86.8** |
| MagMax [41] | 40.2 | 96.1 | 81.1 | 72.0 | 97.8 | 76.3 | 88.4 | 99.2 | 93.0 | 70.5 | 68.9 | 80.3 |
| Mingle-Seq | **58.7** | **97.5** | **87.2** | 79.7 | 97.3 | 87.2 | 90.1 | **99.6** | 93.0 | 80.4 | 73.3 | 85.8 |
| **Continual Merging** (*Independent fine-tuned*) | | | | | | | | | | | | |
| Average (SWA) [26] | 26.5 | 92.3 | 74.3 | 48.4 | 73.7 | 74.0 | 87.1 | 84.0 | 91.2 | 67.5 | 68.5 | 71.6 |
| C. TA [24] | 26.6 | 92.5 | 74.5 | 48.7 | 74.3 | 74.4 | 87.0 | 85.5 | 91.2 | 67.7 | 68.6 | 71.9 |
| C. Ties [86] | 30.5 | 94.0 | 74.8 | 49.8 | 71.7 | 73.8 | 87.3 | 81.5 | 90.6 | 67.0 | 67.9 | 71.7 |
| MagMax-Ind [41] | 29.9 | 93.7 | 78.4 | 46.1 | 58.3 | 68.1 | 86.8 | 82.8 | 91.4 | 62.7 | 69.3 | 69.8 |
| OPCM [70] | 35.7 | 95.9 | 77.0 | 54.6 | 90.3 | 76.4 | 87.1 | 96.3 | 93.3 | 70.1 | 70.5 | 77.0 |
| Mingle | **54.2** | **97.3** | **79.7** | **72.3** | **96.0** | **86.7** | **88.7** | **99.3** | **93.9** | **73.1** | **71.6** | **83.0** |

Table 4: Robustness results of ViT-B/32 continually merged across 4 tasks.

| | Method | Clean | Motion | Impulse | Gaussian | Pixelate | Spatter | Contrast | JPEG | Avg. |
|---|---|---|---|---|---|---|---|---|---|---|
| ACC (%) ↑ | C. LW AdaMerging [87] | 56.0 ±5.3 | 47.5 ±4.4 | 43.1 ±2.3 | 43.3 ±3.4 | 18.1 ±4.7 | 46.6 ±3.0 | 48.9 ±4.8 | 49.1 ±4.0 | 44.9 |
| | C. WEMOE [68] | 3.4 ±0.8 | 3.1 ±0.4 | 4.3 ±1.4 | 3.4 ±1.4 | 3.0 ±1.6 | 4.0 ±0.9 | 3.3 ±0.7 | 4.0 ±1.2 | 3.6 |
| | C. LoRA-WEMOE [68] | 78.7 ±4.5 | 71.0 ±4.9 | 55.0 ±3.8 | 59.4 ±3.8 | 24.9 ±24.9 | 60.5 ±3.8 | 68.5 ±4.8 | 69.7 ±4.4 | 61.0 |
| | C. Task Arithmetic [24] | 77.5 ±0.0 | 66.0 ±0.0 | 58.9 ±0.0 | 59.6 ±0.0 | 29.7 ±0.0 | 63.5 ±0.0 | 66.0 ±0.0 | 67.8 ±0.0 | 61.1 |
| | MagMax-Ind [41] | 79.1 ±0.0 | 69.0 ±0.0 | 60.6 ±0.0 | 61.5 ±0.0 | 33.0 ±0.0 | 66.4 ±0.0 | 68.6 ±0.0 | 69.9 ±0.0 | 63.5 |
| | OPCM [70] | 83.6 ±0.5 | 72.5 ±0.6 | 64.7 ±1.2 | 65.2 ±1.2 | 35.2 ±0.6 | 70.5 ±0.5 | 72.5 ±0.6 | 74.4 ±0.3 | 67.3 |
| | Mingle (Ours) | **89.9** ±0.4 | **82.8** ±0.8 | **67.5** ±2.0 | **70.7** ±1.2 | **37.9** ±0.4 | **77.0** ±0.7 | **80.1** ±0.8 | **82.9** ±0.9 | **73.2** |
| BWT (%) ↑ | C. LW AdaMerging [87] | -38.0 ±7.1 | -37.3 ±5.9 | -22.2 ±3.0 | -25.2 ±4.5 | -20.8 ±6.3 | -28.6 ±4.0 | -34.7 ±6.5 | -36.1 ±5.3 | -29.5 |
| | C. WEMOE [68] | -30.7 ±3.1 | -28.7 ±3.8 | -22.1 ±11.6 | -25.5 ±9.8 | -8.0 ±4.5 | -23.4 ±11.0 | -27.6 ±5.6 | -28.6 ±5.2 | -24.3 |
| | C. LoRA-WEMOE [68] | -13.6 ±6.9 | -14.6 ±8.2 | -11.3 ±4.7 | -10.2 ±3.7 | -15.6 ±8.8 | -10.8 ±3.9 | -11.2 ±8.8 | -16.7 ±6.6 | -13.0 |
| | C. Task Arithmetic [24] | -4.8 ±0.9 | -6.1 ±1.2 | -1.6 ±3.0 | -1.6 ±1.7 | -2.7 ±1.5 | -3.1 ±2.5 | -6.1 ±1.2 | -5.1 ±0.8 | -3.9 |
| | MagMax-Ind [41] | -7.7 ±0.8 | -8.1 ±1.5 | -6.1 ±4.9 | -5.1 ±3.7 | -3.5 ±3.0 | -7.3 ±2.9 | -8.4 ±1.6 | -8.2 ±1.2 | -6.8 |
| | OPCM [70] | -4.3 ±1.8 | -4.5 ±2.8 | -6.4 ±7.1 | -6.1 ±4.3 | -2.9 ±0.9 | -6.3 ±2.9 | -4.5 ±2.8 | -5.7 ±1.5 | -5.1 |
| | Mingle (Ours) | **-0.2** ±0.2 | **-0.1** ±0.4 | **0.7** ±1.0 | **0.6** ±1.1 | **-0.2** ±1.1 | **0.2** ±0.7 | **0.0** ±0.5 | **0.5** ±0.5 | **-0.2** |

consistently delivers state-of-the-art accuracy while nearly eliminating forgetting under diverse continual scenarios.

**Comparison with Conventional CL.** Tab. 3 evaluates two CL paradigms on the MTIL benchmark: conventional CL, where each task model is fine-tuned from its immediate predecessor; and continual merging, which fine-tunes each task model independently before fusion, eliminating inter-model dependencies and enabling flexible task ordering and model reuse. Within the merging family, Mingle sets a new state-of-the-art, and when integrated into a sequential fine-tuning pipeline, it matches the performance of SOTA CL methods. This demonstrates both its strength as a fusion strategy and its versatility across different training regimes.

**Robustness to Test-Time Distribution Shifts.** Following prior work [87, 68], we evaluate Mingle on seven corruptions (motion blur, impulse noise, Gaussian noise, pixelate, spatter, contrast, JPEG) and report results in Tab. 4. It preserves high accuracy and near-zero or even positive BWT, outperforming all baselines, whereas direct application of SOTA TTA-based merging (WEMOE, AdaMerging) in a continual setting fails without tailored designs to continual setup.

## 4.3 Ablation Results and Analysis

**Ablation Study.** We explore the contribution of each component in Tab. 5. Row 1 shows a fixed-weight merging of low-rank experts as our baseline. In Row 2, adding TTA boosts ACC substantially but at the cost of worsening BWT with more tasks. Row 3 demonstrates that freezing earlier gates curbs forgetting while retaining ACC gains. Row 4 then applies null-space constraints, yielding further BWT improvements. Finally, Row 5 presents the full method with adaptive relaxation, which best harmonizes accuracy and long-term stability.

Table 5: Ablation study of MINGLE with CLIP ViT-B/16 over 8, 14, and 20 tasks.

| Test-Time Adaptation | Frozen Old Gate | Null-Space Constrained Gate | Adaptive Relaxation | ACC(%) ↑ | | | BWT(%) ↑ | | |
|---|---|---|---|---|---|---|---|---|---|
| | | | | 8 tasks | 14 tasks | 20 tasks | 8 tasks | 14 tasks | 20 tasks |
| ✗ | – | – | – | $78.7_{\pm0.1}$ | $76.4_{\pm1.0}$ | $70.6_{\pm0.4}$ | $-0.5_{\pm0.1}$ | $-1.0_{\pm0.3}$ | $-1.3_{\pm0.3}$ |
| ✓ | ✗ | ✗ | ✗ | $86.4_{\pm5.3}$ | $81.7_{\pm2.3}$ | $76.7_{\pm1.3}$ | $-6.0_{\pm5.2}$ | $-7.7_{\pm-3.4}$ | $-12.8_{\pm1.3}$ |
| ✓ | ✓ | ✗ | ✗ | $87.4_{\pm0.4}$ | $81.3_{\pm0.8}$ | $76.2_{\pm1.3}$ | $-2.3_{\pm0.5}$ | $-4.3_{\pm0.8}$ | $-6.8_{\pm0.9}$ |
| ✓ | ✓ | ✓ | ✗ | $86.0_{\pm1.5}$ | $83.5_{\pm0.9}$ | $78.3_{\pm1.7}$ | $-0.1_{\pm0.1}$ | $-0.1_{\pm0.1}$ | $-0.2_{\pm0.1}$ |
| ✓ | ✓ | ✓ | ✓ | $88.3_{\pm0.6}$ | $84.9_{\pm0.8}$ | $81.9_{\pm0.9}$ | $-0.4_{\pm0.1}$ | $-0.9_{\pm0.1}$ | $-1.9_{\pm0.4}$ |

Table 6: Ablation on number of adaptation steps for ViT-B/32 across 8, 14, and 20 tasks.

| Steps | ACC (8-task) | BWT (8-task) | ACC (14-task) | BWT (14-task) | ACC (20-task) | BWT (20-task) |
|---|---|---|---|---|---|---|
| 5 | $60.9_{\pm1.4}$ | $-0.1_{\pm0.2}$ | $62.4_{\pm1.7}$ | $-0.3_{\pm0.1}$ | $60.2_{\pm1.5}$ | $-0.1_{\pm0.2}$ |
| 10 | $68.6_{\pm1.6}$ | $-0.2_{\pm0.2}$ | $68.5_{\pm1.6}$ | $-0.1_{\pm0.2}$ | $63.5_{\pm1.0}$ | $-0.4_{\pm0.3}$ |
| 20 | $78.4_{\pm0.6}$ | $-0.2_{\pm0.1}$ | $75.5_{\pm1.3}$ | $-0.4_{\pm0.1}$ | $71.0_{\pm1.2}$ | $-0.4_{\pm0.4}$ |
| 50 | $85.8_{\pm0.8}$ | $-0.6_{\pm0.4}$ | $81.6_{\pm1.4}$ | $-1.1_{\pm0.3}$ | $77.1_{\pm2.0}$ | $-2.2_{\pm0.8}$ |

Table 7: Efficiency and sample analysis. (a) Expert insertion layers and rank sweep over 8 tasks on CLIP-ViT-B/32. (b) Wall-clock adaptation time across tasks on CLIP-ViT-B/32. (c) Accuracy (%) of CLIP-ViT-B/16 under varying numbers of test samples.

**(a) Expert insertion layers and rank sweep (8 tasks)**

| Configuration | TTA Time | Train. Param | Full Param | ACC(%) |
|---|---|---|---|---|
| attn.qkv_proj ($r = 64$) | 61 s | 27.7 k | 116.0 M | 69.9 |
| attn.out_proj ($r = 64$) | 47 s | 9.3 k | 97.0 M | 53.9 |
| mlp.fc1 ($r = 64$) | 48 s | 9.0 k | 111.1 M | 82.6 |
| mlp.fc2 ($r = 64$) | 48 s | 36.8 k | 111.3 M | 70.1 |
| attn & mlp ($r = 64$) | 78 s | 83.0 k | 173.1 M | 85.8 |
| qkv & fc1 ($r = 32$) | 65 s | 36.9 k | 113.7 M | 84.5 |
| qkv & fc1 ($r = 128$) | 65 s | 36.9 k | 191.6 M | 85.1 |
| qkv & fc1 ($r = 768$) | 70 s | 36.9 k | 710.3 M | 83.7 |
| qkv & fc1 ($r = 64$) | 65 s | 36.9 k | 139.7 M | 85.0 |

**(b) Wall-clock adaptation time across tasks**

| #Tasks | Adaptation steps | Total Time (s) | Avg./task (s) |
|---|---|---|---|
| 8 | 50 | 78 | 9.8 |
| 14 | 50 | 138 | 9.9 |
| 20 | 50 | 211 | 10.6 |

**(c) Results with varying samples/class**

| # Samples/class | 8-task | 14-task | 20-task |
|---|---|---|---|
| 0 (Static) | $78.7_{\pm0.1}$ | $76.4_{\pm1.0}$ | $70.6_{\pm0.4}$ |
| 1 | $88.4_{\pm0.4}$ | $84.6_{\pm1.1}$ | $81.7_{\pm1.9}$ |
| 3 | $88.6_{\pm0.4}$ | $84.7_{\pm1.0}$ | $81.7_{\pm1.0}$ |
| 5 | $88.3_{\pm0.7}$ | $84.9_{\pm0.8}$ | $81.9_{\pm0.9}$ |
| 10 | $88.6_{\pm0.3}$ | $85.1_{\pm1.1}$ | $82.1_{\pm1.2}$ |

**Ablation on TTA optimization steps.** Tab. 6 reports accuracy and backward transfer of CLIP ViT-B/32 under different adaptation steps across 8, 14, and 20 tasks. Accuracy improves steadily with longer schedules, rising from 60–63% at 5 steps to 77–86% at 50 steps. Forgetting remains negligible, with BWT close to zero in all cases and only a minor drop of about 2% in the 20-task setting at 50 steps. Notably, as few as 20 steps are sufficient to surpass all baselines in Tab. 1, while 50 steps yield the best accuracy with only a modest increase in cost. These results confirm that MINGLE is both effective under tight compute budgets and scalable with additional adaptation.

**Computation and Parameter Efficiency.** Tab. 7 summarizes the efficiency analysis. In part (a), inserting experts into both attn and mlp layers yields the highest accuracy, 85.8%, but also the longest adaptation time of 78s and 83k trainable parameters. A lighter hybrid scheme qkv & fc1 with rank 64 reaches 85.0% accuracy with 36.9k parameters and 65s, offering a better trade-off. The rank sweep shows that raising the rank from 32 to 64 improves accuracy from 84.5% to 85.0%, while larger ranks bring little or even negative gain, *e.g.*, rank 768 drops to 83.7%. Part (b) reports wall-clock adaptation time as tasks increase: with 20 tasks the total is 211s, averaging about 10s per task. After adaptation the router remains fixed and inference is purely feedforward without TTA, enabling low-latency deployment across all tasks. Overall, MINGLE achieves a strong balance of accuracy, parameter efficiency, and scalability under diverse resource budgets.

**Number of Seed Samples.** The number of seed samples per class is crucial for TTA reliability and efficiency. As shown in Tab. 7 (c), using no samples reduces to the *Static* baseline, where LoRA modules are merged with fixed coefficients (0.3) without adaptation, yielding 70–79% accuracy. Introducing a single sample lifts accuracy to 81–88%, and adding more samples offers only minor gains. Variance across task orders decreases with more samples, making five samples per class a balanced trade-off between performance and efficiency.

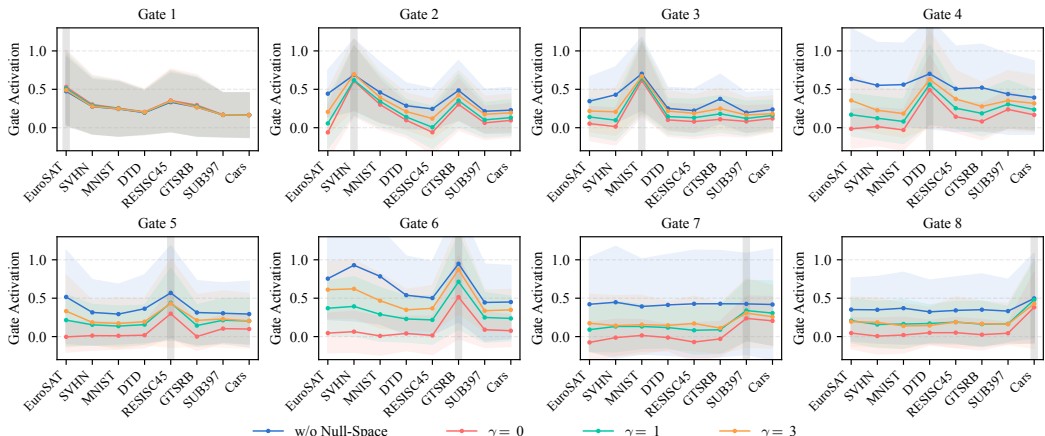

Figure 3: Gate activations across eight tasks under varying $\gamma$. Each subplot shows one gate; curves and shaded areas indicate mean and std across layers. Gray bars mark the gate's training task. Lower $\gamma$ leads to stronger suppression on prior tasks.

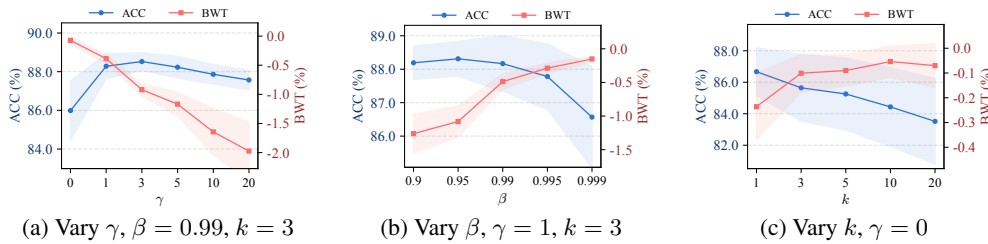

(a) Vary $\gamma$, $\beta = 0.99$, $k = 3$     (b) Vary $\beta$, $\gamma = 1$, $k = 3$     (c) Vary $k$, $\gamma = 0$

Figure 4: Sensitivity analysis of the null-space constrained gating *w.r.t.* hyper-parameters $\beta$, $\gamma$, and $k$.

**Visualization on Gate Activations.** Fig. 3 shows that the null-space constraint suppresses gate responses on previously seen tasks, reducing forgetting, with smaller $\gamma$ giving stronger attenuation. We also observe that gate activations remain below 1.0 across tasks, even in the w/o Null-Space variant (blue curve), showing that under-activation is not solely due to the constraint. Instead, it reflects the complementary nature of experts: multiple LoRA modules capture overlapping but distinct subspaces, and softly combining them often yields better performance, especially in settings like TTCMM where task boundaries are fuzzy.

**Hyper-parameter of Gate.** We study the effect of three key hyper-parameters: $\gamma$, $\beta$, and $k$. $\gamma$ controls the strength of null-space suppression; smaller values lead to stronger attenuation of activations on prior tasks, reducing forgetting (Fig. 4a). As shown in Fig. 4b, $\beta$ regulates the smoothness of the EMA used to accumulate interference signals. A moderate setting ($\beta = 0.99$) balances responsiveness and stability. Smaller $\beta$ amplifies noise sensitivity, while larger $\beta$ slows detection of interference. $k$ determines the number of principal directions retained per task. The mitigation of forgetting saturates at $k = 3$ (Fig. 4c), indicating that a small number of task-specific directions suffices.

## 5 Conclusions

In this work, we introduced the task of test-time continual model merging (TTCMM) and proposed MINGLE, a novel framework for TTCMM that integrates a mixture-of-experts architecture with adaptive null-space constrained gating. Extensive empirical evaluations show that MINGLE substantially improves generalization and mitigates catastrophic forgetting, consistently outperforming prior state-of-the-art approaches. These results establish TTCMM as a principled paradigm for addressing both task interference and distribution shift, and highlight the practical potential of MINGLE for scalable and efficient continual learning in real-world applications.

**Acknowledgments**    This work was supported in part by National Science and Technology Major Project (2021ZD0112001), National Natural Science Foundation of China (No.62271119, 08120002, 62071086, U23A20286), the Key Research and Development Project of Hainan Province (Grant No. ZDYF2024(LALH)003), the Fundamental Research Funds for the Central University of China (DUT No. 82232031), the Natural Science Foundation of Sichuan Province under Grant 2025ZNSFSC0475.

We thank all reviewers for taking the time to review our paper and give valuable suggestions.

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

# Appendix

## A  Theoretical Risk Comparison: Dynamic MoE vs. Static Averaging

**Problem Setup and Definitions**  Consider $T$ independent tasks, each associated with a data distribution $D_t$ for $t = 1, \ldots, T$. For each task $t$, a pre-trained expert model $f_t(x)$ outputs a probability distribution over classes, trained specifically on $D_t$. The overall data distribution $D$ is a mixture of these tasks, where an example $(x, y)$ is drawn from task $t$ with prior probability $P(t)$, and then $(x, y) \sim D_t$. The expected risk of a predictive model $h(x)$ is defined as:

$$R(h) = \mathbb{E}_{(x,y)\sim D}\big[\ell(h(x), y)\big] = \sum_{t=1}^{T} P(t)\,\mathbb{E}_{(x,y)\sim D_t}\big[\ell(h(x), y)\big], \tag{12}$$

where $\ell(h(x), y)$ is a loss function (*e.g.*, cross-entropy or 0-1 loss).

We compare two methods to combine the experts into a final prediction $h(x)$:

- **Static Averaging**: Defined as $h_{\text{static}}(x) = \sum_{i=1}^{T} \alpha_i f_i(x)$, where $\boldsymbol{\alpha} = (\alpha_1, \ldots, \alpha_T)$ is a fixed weight vector independent of $x$, typically with $\alpha_i \geq 0$ and $\sum_i \alpha_i = 1$ for probability outputs.

- **Dynamic Mixture-of-Experts (MoE)**: Defined as $h_{\text{MoE}}(x) = f_{i^*(x)}(x)$, where $i^*(x) = \arg\max_i g_i(x)$ and $g(x) = (g_1(x), \ldots, g_T(x))$ is a gating function that selects one expert per input (hard routing). The gating is subject to routing noise, modeled below.

**Routing Noise Model**  For each input $x$ drawn from $D_t$, the true task is $t$, and the ideal expert is $f_t$. The gating selects the correct expert $i^*(x) = t$ with probability $1 - \varepsilon_t$, and an incorrect expert $i^*(x) \neq t$ with probability $\varepsilon_t = P(i^*(x) \neq t \mid x \sim D_t)$, the task-specific routing error rate. On error, the gating selects a random expert from $\{1, \ldots, T\} \setminus \{t\}$ uniformly. Define:

- $R_t(i) = \mathbb{E}_{(x,y)\sim D_t}[\ell(f_i(x), y)]$, the risk of expert $i$ on task $t$.

- $R_{\text{ideal}} = \sum_{t=1}^{T} P(t) R_t(t)$, the risk with perfect routing.

- $R_{\text{wrong},t} = \frac{1}{T-1} \sum_{i \neq t} R_t(i)$, the average risk of incorrect experts on task $t$.

- $\varepsilon = \sum_{t=1}^{T} P(t)\varepsilon_t$, the overall routing error rate.

**Theorem A.1** (Dynamic MoE versus Static Averaging). *Let $\{(D_t, f_t)\}_{t=1}^T$ be $T$ independent tasks with priors $P(t)$ and per-task risks $R_t(i)$. For any static mixture $h_{\text{static}}(x) = \sum_{i=1}^T \alpha_i f_i(x)$ and any hard-routed MoE $h_{\text{MoE}}(x) = f_{i^\star(x)}(x)$ with task-specific routing errors $\varepsilon_t$:*

$$R(h_{\text{MoE}}) \;=\; R_{\text{ideal}} + \sum_{t=1}^T P(t)\,\varepsilon_t\big(R_{wrong,t} - R_t(t)\big), \tag{13}$$

*where $R_{\text{ideal}} = \sum_t P(t)R_t(t)$ and $R_{wrong,t} = \frac{1}{T-1}\sum_{i \neq t} R_t(i)$. Moreover,*

1. *(Perfect routing) If $\varepsilon_t = 0$ for all $t$, then $\inf_g R(h_{\text{MoE}}) \;<\; \inf_\alpha R(h_{\text{static}})$ whenever at least two tasks disagree on their best expert.*

2. *(Noisy routing) If $\sum_t P(t)\varepsilon_t\big(R_{wrong,t} - R_t(t)\big) \;<\; R^*_{static} - R_{\text{ideal}}$, where $R^*_{static} = \inf_\alpha R(h_{static})$, then the MoE still attains lower risk than any static mixture.*

*Proof.* The proof proceeds in three parts: (1) deriving the MoE risk with routing noise, (2) proving the optimal gating case, and (3) establishing the condition for MoE superiority under routing noise.

**Step 1: MoE Risk with Routing Noise**

The MoE prediction is $h_{\text{MoE}}(x) = f_{i^*(x)}(x)$, where $i^*(x) = \arg\max_i g_i(x)$. The expected risk is:

$$R(h_{\text{MoE}}) = \mathbb{E}_{(x,y)\sim D}[\ell(f_{i^*(x)}(x), y)] = \sum_{t=1}^T P(t)\,\mathbb{E}_{(x,y)\sim D_t}[\ell(f_{i^*(x)}(x), y)]. \tag{14}$$

For task $t$, condition on routing correctness:

- **Correct routing** ($i^*(x) = t$): Probability $1 - \varepsilon_t$, risk $R_t(t)$.

- **Incorrect routing** ($i^*(x) \neq t$): Probability $\varepsilon_t$, selects a random expert from $\{1, \ldots, T\} \setminus \{t\}$, with average risk $R_{\text{wrong},t} = \frac{1}{T-1}\sum_{i \neq t} R_t(i)$.

The expected risk on task $t$ is:

$$\mathbb{E}_{(x,y)\sim D_t}[\ell(f_{i^*(x)}(x), y)] = (1 - \varepsilon_t)R_t(t) + \varepsilon_t R_{\text{wrong},t}. \tag{15}$$

Thus, the total risk is:

$$R(h_{\text{MoE}}) = \sum_{t=1}^T P(t)\big[(1 - \varepsilon_t)R_t(t) + \varepsilon_t R_{\text{wrong},t}\big]. \tag{16}$$

Rewrite:

$$R(h_{\text{MoE}}) = \sum_{t=1}^T P(t)R_t(t) + \sum_{t=1}^T P(t)\varepsilon_t(R_{\text{wrong},t} - R_t(t)) = R_{\text{ideal}} + \sum_{t=1}^T P(t)\varepsilon_t\delta_t, \tag{17}$$

where $\delta_t = R_{\text{wrong},t} - R_t(t) > 0$ is the risk increase due to misrouting on task $t$.

**Step 2: Optimal Gating (No Routing Noise)**

Assume an oracle gating function with $\varepsilon_t = 0$ for all $t$, so $i^*(x) = t$ for all $x \sim D_t$. Then:

$$R(h_{\text{MoE}}) = \sum_{t=1}^T P(t)R_t(t) = R_{\text{ideal}}. \tag{18}$$

Define hypothesis classes:

$$\mathcal{H}_{\text{static}} = \left\{ x \mapsto \sum_{i=1}^T \alpha_i f_i(x) \mid \boldsymbol{\alpha} \in \mathbb{R}^T \right\}, \tag{19}$$

$$\mathcal{H}_{\text{MoE}} = \left\{ x \mapsto f_{i^*(x)}(x) \mid i^*(x) = \arg\max_i g_i(x), g : \mathcal{X} \to \mathbb{R}^T \right\}. \tag{20}$$

Any static model $h_{\text{static}}(x) = \sum_{i=1}^T \alpha_i f_i(x)$ can be approximated by an MoE with $g(x)$ assigning constant weights, so $\mathcal{H}_{\text{static}} \subseteq \mathcal{H}_{\text{MoE}}$. Thus:

$$R^*_{\text{MoE}} = \inf_{g(\cdot)} R(h_{\text{MoE}}) \leq \inf_{\boldsymbol{\alpha}} R(h_{\text{static}}) = R^*_{\text{static}}. \tag{21}$$

Under task heterogeneity ($R_t(t) < R_t(s)$ and $R_s(s) < R_s(t)$ for some $t \neq s$), the ideal MoE routes each $x \sim D_t$ to $f_t$, achieving:

$$R_{\text{ideal}} = \sum_{t=1}^T P(t) R_t(t). \tag{22}$$

For static averaging:

$$R(h_{\text{static}}) = \sum_{t=1}^T P(t) \mathbb{E}_{(x,y) \sim D_t} \left[ \ell \left( \sum_{i=1}^T \alpha_i f_i(x), y \right) \right]. \tag{23}$$

Since $\ell$ is convex (*e.g.*, cross-entropy), Jensen's inequality implies:

$$\mathbb{E}_{D_t} \left[ \ell \left( \sum_i \alpha_i f_i(x), y \right) \right] \geq R_t(t), \tag{24}$$

with strict inequality unless $\alpha_t = 1$ and $\alpha_i = 0$ for $i \neq t$, which cannot hold for all tasks simultaneously under heterogeneity. Thus:

$$R^*_{\text{static}} > R_{\text{ideal}} = R^*_{\text{MoE}}. \tag{25}$$

**Step 3: MoE Superiority with Routing Noise**

Let $\gamma = R^*_{\text{static}} - R_{\text{ideal}} > 0$ under task heterogeneity. The MoE outperforms the static model if:

$$R(h_{\text{MoE}}) < R^*_{\text{static}}. \tag{26}$$

Substitute:

$$R_{\text{ideal}} + \sum_{t=1}^T P(t) \varepsilon_t \delta_t < R_{\text{ideal}} + \gamma. \tag{27}$$

Thus:

$$\sum_{t=1}^T P(t) \varepsilon_t \delta_t < \gamma = R^*_{\text{static}} - R_{\text{ideal}}. \tag{28}$$

Since $\delta_t = R_{\text{wrong},t} - R_t(t)$, the condition is:

$$\sum_{t=1}^T P(t) \varepsilon_t (R_{\text{wrong},t} - R_t(t)) < R^*_{\text{static}} - R_{\text{ideal}}. \tag{29}$$

If this holds, the MoE's risk, despite routing noise, remains below the best static risk. $\square$

**Conclusion**: The MoE risk is $R_{\text{ideal}} + \sum_{t=1}^T P(t) \varepsilon_t (R_{\text{wrong},t} - R_t(t))$, and it outperforms static averaging when routing noise is sufficiently small relative to the static model's suboptimality. The optimal gating case confirms $R^*_{\text{MoE}} \leq R^*_{\text{static}}$, with strict inequality under task heterogeneity.

## B  Additional Descriptions

### B.1  Details of Dataset and Task Settings

**Dataset Details**    Following prior works [70], we evaluate continual model merging on twenty publicly available image classification datasets, including SUN397 [84], Stanford Cars [32], RESISC45 [4], EuroSAT [19], SVHN [46], GTSRB [64], MNIST [34], DTD [7], Flowers102 [47], PCAM [73], FER2013 [18], Oxford-IIIT Pet [50], STL-10 [9], CIFAR-100 and CIFAR-10 [33], Food-101 [3], Fashion-MNIST [83], EMNIST [10], KMNIST [8], and Rendered SST-2 [62].

Table 8: Extended downstream datasets used in our experiments.

| Dataset | #Classes | #Train (k) | #Test (k) | Task |
|---|---|---|---|---|
| SUN397 | 287 | 19.9 | 19.9 | Scene category |
| Stanford Cars | 196 | 8.1 | 8.0 | Car series |
| RESISC45 | 45 | 18.9 | 6.3 | Remote–sensing scene |
| EuroSAT | 10 | 21.6 | 2.7 | Satellite land-use |
| SVHN | 10 | 73.3 | 26.0 | Digit recognition |
| GTSRB | 43 | 39.2 | 12.6 | Traffic sign |
| MNIST | 10 | 60 | 10 | Hand-written digit |
| DTD | 47 | 3.8 | 1.9 | Texture recognition |
| Flowers102 | 102 | 1.0 | 6.1 | Flower species |
| PCAM | 2 | 262 | 32.8 | Tumour detection |
| FER2013 | 7 | 28.7 | 3.6 | Facial emotion |
| Oxford IIIT Pet | 37 | 3.7 | 3.7 | Animal species |
| STL10 | 10 | 5 | 8 | Object recognition |
| CIFAR-100 | 100 | 50 | 10 | Natural object |
| CIFAR-10 | 10 | 50 | 10 | Natural object |
| Food101 | 101 | 75.8 | 25.3 | Food type |
| Fashion-MNIST | 10 | 60 | 10 | Fashion product |
| EMNIST (digits) | 10 | 60 | 10 | Hand-written digit |
| KMNIST | 10 | 60 | 10 | Kuzushiji character |
| Rendered SST-2 | 2 | 6.9 | 1.8 | Rendered sentiment |

**Task Grouping** We group the 20 datasets into three progressive task sets and evaluate the merged models using average accuracy (ACC) and backward transfer (BWT) metrics. For each task group, we perform 10 experiments using different task sequences (listed in Tab. 9), and report both the mean and standard deviation of the results to ensure robustness and consistency.

- **8-task group**: (1) SUN397, (2) Stanford Cars, (3) RESISC45, (4) EuroSAT, (5) SVHN, (6) GTSRB, (7) MNIST, (8) DTD.

- **14-task group**: (1) SUN397, (2) Stanford Cars, (3) RESISC45, (4) EuroSAT, (5) SVHN, (6) GTSRB, (7) MNIST, (8) DTD, (9) Flowers102, (10) PCAM, (11) FER2013, (12) OxfordIIITPet, (13) STL10, (14) CIFAR100.

- **20-task group**: (1) SUN397, (2) Stanford Cars, (3) RESISC45, (4) EuroSAT, (5) SVHN, (6) GTSRB, (7) MNIST, (8) DTD, (9) Flowers102, (10) PCAM, (11) FER2013, (12) OxfordIIITPet, (13) STL10, (14) CIFAR100, (15) CIFAR10, (16) Food101, (17) FashionMNIST, (18) EMNIST, (19) KMNIST, (20) RenderedSST2.

## B.2 Details of Downstream Models

In this section, we present the evaluation setup for pre-trained and fine-tuned models. As shown in Tab. 10, we evaluate the zero-shot accuracy of the original CLIP-ViT models and the performance of fine-tuned models on the test sets of various downstream tasks. The fine-tuned checkpoints are obtained directly from Hugging Face (`https://huggingface.co/tanganke`), where each model has been fine-tuned on task-specific training data using a standard protocol. The visual encoder is updated during fine-tuning, while the classification head is fixed and initialized from the pre-trained text encoder. The fine-tuning setup follows a standard configuration: cross-entropy loss, Adam optimizer, cosine annealing learning rate schedule with a peak learning rate of 1e-5, batch size 128, and 4000 training steps.

## B.3 Details of Baselines

Our experiments involve the following comparison methods and our method:

- **Stochastic Weight Averaging (SWA).** A simple model averaging technique to stabilize optimization and improve generalization [26]. At each step $t$, the model parameters are averaged across previous checkpoint: $\theta_t^{\text{SWA}} = \frac{1}{t} \left[ \theta_{t-1}^{\text{SWA}}(t-1) + \theta_t^{\text{SWA}} \right]$. This approach can be interpreted

Table 9: Dataset orderings used for experiments in each task group.

| Group | Order | Dataset Order (by ID) |
|-------|-------|------------------------|
| 8 tasks | 1 | $(04 \rightarrow 05 \rightarrow 07 \rightarrow 08 \rightarrow 03 \rightarrow 06 \rightarrow 01 \rightarrow 02)$ |
| | 2 | $(07 \rightarrow 08 \rightarrow 05 \rightarrow 04 \rightarrow 02 \rightarrow 06 \rightarrow 03 \rightarrow 01)$ |
| | 3 | $(03 \rightarrow 06 \rightarrow 04 \rightarrow 02 \rightarrow 01 \rightarrow 08 \rightarrow 05 \rightarrow 07)$ |
| | 4 | $(06 \rightarrow 08 \rightarrow 02 \rightarrow 01 \rightarrow 03 \rightarrow 07 \rightarrow 04 \rightarrow 05)$ |
| | 5 | $(07 \rightarrow 06 \rightarrow 03 \rightarrow 08 \rightarrow 05 \rightarrow 01 \rightarrow 04 \rightarrow 02)$ |
| | 6 | $(07 \rightarrow 02 \rightarrow 03 \rightarrow 08 \rightarrow 05 \rightarrow 04 \rightarrow 01 \rightarrow 06)$ |
| | 7 | $(07 \rightarrow 01 \rightarrow 04 \rightarrow 03 \rightarrow 08 \rightarrow 05 \rightarrow 02 \rightarrow 06)$ |
| | 8 | $(08 \rightarrow 05 \rightarrow 06 \rightarrow 07 \rightarrow 01 \rightarrow 04 \rightarrow 03 \rightarrow 02)$ |
| | 9 | $(01 \rightarrow 04 \rightarrow 05 \rightarrow 02 \rightarrow 06 \rightarrow 03 \rightarrow 07 \rightarrow 08)$ |
| | 10 | $(08 \rightarrow 03 \rightarrow 01 \rightarrow 02 \rightarrow 06 \rightarrow 05 \rightarrow 07 \rightarrow 04)$ |
| 14 tasks | 1 | $(09 \rightarrow 13 \rightarrow 08 \rightarrow 07 \rightarrow 14 \rightarrow 12 \rightarrow 06 \rightarrow 03 \rightarrow 10 \rightarrow 04 \rightarrow 05 \rightarrow 01 \rightarrow 02 \rightarrow 11)$ |
| | 2 | $(09 \rightarrow 10 \rightarrow 11 \rightarrow 14 \rightarrow 07 \rightarrow 13 \rightarrow 04 \rightarrow 02 \rightarrow 06 \rightarrow 08 \rightarrow 03 \rightarrow 12 \rightarrow 05 \rightarrow 01)$ |
| | 3 | $(05 \rightarrow 08 \rightarrow 12 \rightarrow 06 \rightarrow 11 \rightarrow 01 \rightarrow 10 \rightarrow 04 \rightarrow 14 \rightarrow 03 \rightarrow 02 \rightarrow 13 \rightarrow 09 \rightarrow 07)$ |
| | 4 | $(03 \rightarrow 10 \rightarrow 09 \rightarrow 12 \rightarrow 04 \rightarrow 13 \rightarrow 01 \rightarrow 06 \rightarrow 11 \rightarrow 02 \rightarrow 14 \rightarrow 08 \rightarrow 07 \rightarrow 05)$ |
| | 5 | $(08 \rightarrow 14 \rightarrow 09 \rightarrow 06 \rightarrow 12 \rightarrow 13 \rightarrow 05 \rightarrow 03 \rightarrow 04 \rightarrow 11 \rightarrow 10 \rightarrow 01 \rightarrow 07 \rightarrow 02)$ |
| | 6 | $(03 \rightarrow 12 \rightarrow 13 \rightarrow 01 \rightarrow 11 \rightarrow 04 \rightarrow 10 \rightarrow 05 \rightarrow 14 \rightarrow 08 \rightarrow 09 \rightarrow 07 \rightarrow 02 \rightarrow 06)$ |
| | 7 | $(07 \rightarrow 01 \rightarrow 12 \rightarrow 10 \rightarrow 02 \rightarrow 08 \rightarrow 13 \rightarrow 04 \rightarrow 05 \rightarrow 11 \rightarrow 14 \rightarrow 03 \rightarrow 06 \rightarrow 09)$ |
| | 8 | $(05 \rightarrow 12 \rightarrow 04 \rightarrow 11 \rightarrow 03 \rightarrow 08 \rightarrow 10 \rightarrow 01 \rightarrow 09 \rightarrow 13 \rightarrow 14 \rightarrow 07 \rightarrow 06 \rightarrow 02)$ |
| | 9 | $(10 \rightarrow 07 \rightarrow 09 \rightarrow 02 \rightarrow 03 \rightarrow 13 \rightarrow 01 \rightarrow 12 \rightarrow 14 \rightarrow 04 \rightarrow 11 \rightarrow 06 \rightarrow 05 \rightarrow 08)$ |
| | 10 | $(01 \rightarrow 02 \rightarrow 11 \rightarrow 06 \rightarrow 08 \rightarrow 12 \rightarrow 07 \rightarrow 05 \rightarrow 10 \rightarrow 14 \rightarrow 03 \rightarrow 13 \rightarrow 09 \rightarrow 04)$ |
| 20 tasks | 1 | $(20 \rightarrow 06 \rightarrow 15 \rightarrow 05 \rightarrow 10 \rightarrow 14 \rightarrow 16 \rightarrow 19 \rightarrow 07 \rightarrow 13 \rightarrow 18 \rightarrow 11 \rightarrow 02 \rightarrow 12 \rightarrow 03 \rightarrow 17 \rightarrow 08 \rightarrow 09 \rightarrow 01 \rightarrow 04)$ |
| | 2 | $(09 \rightarrow 14 \rightarrow 06 \rightarrow 03 \rightarrow 07 \rightarrow 04 \rightarrow 18 \rightarrow 01 \rightarrow 17 \rightarrow 19 \rightarrow 08 \rightarrow 20 \rightarrow 13 \rightarrow 16 \rightarrow 11 \rightarrow 12 \rightarrow 15 \rightarrow 05 \rightarrow 10 \rightarrow 02)$ |
| | 3 | $(09 \rightarrow 15 \rightarrow 16 \rightarrow 11 \rightarrow 03 \rightarrow 13 \rightarrow 08 \rightarrow 10 \rightarrow 12 \rightarrow 02 \rightarrow 20 \rightarrow 01 \rightarrow 05 \rightarrow 19 \rightarrow 07 \rightarrow 06 \rightarrow 04 \rightarrow 18 \rightarrow 17 \rightarrow 14)$ |
| | 4 | $(17 \rightarrow 04 \rightarrow 11 \rightarrow 19 \rightarrow 18 \rightarrow 10 \rightarrow 07 \rightarrow 15 \rightarrow 12 \rightarrow 13 \rightarrow 08 \rightarrow 02 \rightarrow 01 \rightarrow 06 \rightarrow 05 \rightarrow 03 \rightarrow 20 \rightarrow 16 \rightarrow 14 \rightarrow 09)$ |
| | 5 | $(14 \rightarrow 16 \rightarrow 04 \rightarrow 20 \rightarrow 15 \rightarrow 17 \rightarrow 07 \rightarrow 11 \rightarrow 06 \rightarrow 18 \rightarrow 12 \rightarrow 01 \rightarrow 19 \rightarrow 09 \rightarrow 10 \rightarrow 05 \rightarrow 08 \rightarrow 02 \rightarrow 13 \rightarrow 03)$ |
| | 6 | $(02 \rightarrow 06 \rightarrow 17 \rightarrow 04 \rightarrow 19 \rightarrow 18 \rightarrow 08 \rightarrow 16 \rightarrow 20 \rightarrow 01 \rightarrow 10 \rightarrow 13 \rightarrow 07 \rightarrow 09 \rightarrow 05 \rightarrow 11 \rightarrow 15 \rightarrow 14 \rightarrow 03 \rightarrow 12)$ |
| | 7 | $(19 \rightarrow 01 \rightarrow 09 \rightarrow 14 \rightarrow 06 \rightarrow 20 \rightarrow 17 \rightarrow 04 \rightarrow 08 \rightarrow 02 \rightarrow 15 \rightarrow 03 \rightarrow 16 \rightarrow 13 \rightarrow 12 \rightarrow 07 \rightarrow 10 \rightarrow 05 \rightarrow 11 \rightarrow 18)$ |
| | 8 | $(15 \rightarrow 07 \rightarrow 08 \rightarrow 02 \rightarrow 10 \rightarrow 06 \rightarrow 17 \rightarrow 20 \rightarrow 05 \rightarrow 19 \rightarrow 16 \rightarrow 01 \rightarrow 18 \rightarrow 09 \rightarrow 13 \rightarrow 11 \rightarrow 04 \rightarrow 14 \rightarrow 12 \rightarrow 03)$ |
| | 9 | $(10 \rightarrow 05 \rightarrow 07 \rightarrow 11 \rightarrow 01 \rightarrow 03 \rightarrow 17 \rightarrow 15 \rightarrow 18 \rightarrow 04 \rightarrow 14 \rightarrow 19 \rightarrow 02 \rightarrow 06 \rightarrow 13 \rightarrow 20 \rightarrow 08 \rightarrow 12 \rightarrow 09 \rightarrow 16)$ |
| | 10 | $(01 \rightarrow 11 \rightarrow 02 \rightarrow 15 \rightarrow 03 \rightarrow 10 \rightarrow 12 \rightarrow 19 \rightarrow 16 \rightarrow 13 \rightarrow 07 \rightarrow 05 \rightarrow 09 \rightarrow 04 \rightarrow 14 \rightarrow 20 \rightarrow 06 \rightarrow 18 \rightarrow 17 \rightarrow 08)$ |

as a form of uniform model ensembling. While conceptually straightforward, SWA treats all checkpoints equally and does not account for inter-task conflicts.

- **Continual Task Arithmetic (C. TA).** A training-free merging strategy that linearly combines task-specific fine-tuned models with a shared pre-trained model [24]. It computes the merged parameters as $\theta_t^{\text{merged}} = \theta_{t-1}^{\text{merged}} + \lambda(\theta_t - \theta_0)$, where $\lambda$ is a scaling factor. TA is computationally efficient and easy to apply, but sensitive to $\lambda$ and prone to destructive interference when merging dissimilar tasks.

- **Continual Ties-Merging (C. Ties).** An extension of Task Arithmetic that reduces parameter-level redundancy and sign conflicts during model merging [86]. For task $t$, the difference vector $\Delta\theta_t = \theta_t - \theta_0$ is trimmed and sign-normalized to obtain $\Delta\theta_t^{\text{Ties}} = \text{Ties}\big(\Delta\theta_{t-1}^{\text{Ties}}, \Delta\theta_t\big)$, and the merged model is given by $\theta_t^{\text{merged}} = \theta_{t-1}^{\text{merged}} + \lambda\Delta\theta_t^{\text{Ties}}$.

- **Orthogonal Projection-based Continual Merging (OPCM).** A projection-based scheme to mitigate task interference by enforcing orthogonality between parameter updates [70]. Specifically, each $\Delta\theta_t$ is projected onto the orthogonal complement of the subspace spanned by previous updates: $\theta_t^{\text{merged}} = \theta_0 + \frac{1}{\lambda_t}\left[\lambda_{t-1}\Delta\theta_{t-1}^{\text{merged}} + \mathcal{P}^{(t-1)}(\Delta\theta_t)\right]$, where $\mathcal{P}^{(t-1)}$ denotes the orthogonal projection.

- **Maximum Magnitude Selection (MagMax).** An extension of Task Arithmetic that, for each parameter dimension, selects the update with the larger absolute value: $\Delta\theta_t^{\text{MagMax}} = \text{MagMax}\big(\Delta\theta_{t-1}^{\text{MagMax}}, \Delta\theta_t\big)$, and the merged model is given by $\theta_t^{\text{merged}} = \theta_{t-1}^{\text{merged}} + \lambda\Delta\theta_t^{\text{MagMax}}$.

Table 10: Test set accuracy of the pre-trained model and individual fine-tuned models on different downstream tasks.

| Model | SUN397 | Cars | RESISC45 | EuroSAT | SVHN | GTSRB | MNIST | DTD | Flowers102 | PCAM |
|---|---|---|---|---|---|---|---|---|---|---|
| **CLIP-ViT-B/32** | | | | | | | | | | |
| Pre-trained | 63.2 | 59.6 | 60.3 | 45.0 | 31.6 | 32.5 | 48.3 | 44.2 | 66.4 | 60.6 |
| Fine-tuned | 74.9 | 78.5 | 95.1 | 99.1 | 97.3 | 98.9 | 99.6 | 79.7 | 88.6 | 88.0 |
| **CLIP-ViT-B/16** | | | | | | | | | | |
| Pre-trained | 65.5 | 64.7 | 66.4 | 54.1 | 52.0 | 43.5 | 51.7 | 45.0 | 71.3 | 54.0 |
| Fine-tuned | 78.9 | 85.9 | 96.6 | 99.0 | 97.6 | 99.0 | 99.7 | 82.3 | 94.9 | 90.6 |
| **CLIP-ViT-L/14** | | | | | | | | | | |
| Pre-trained | 68.2 | 77.9 | 71.3 | 61.2 | 58.4 | 50.5 | 76.3 | 55.5 | 79.2 | 51.2 |
| Fine-tuned | 82.8 | 92.8 | 97.4 | 99.1 | 97.9 | 99.2 | 99.8 | 85.5 | 97.7 | 91.1 |

| Model | FER2013 | OxfordIIITPet | STL10 | CIFAR100 | CIFAR10 | Food101 | FashionMNIST | EMNIST | KMNIST | RenderedSST2 |
|---|---|---|---|---|---|---|---|---|---|---|
| **CLIP-ViT-B/32** | | | | | | | | | | |
| Pre-trained | 41.3 | 83.3 | 97.1 | 63.7 | 89.8 | 82.4 | 63.0 | 12.0 | 10.0 | 58.6 |
| Fine-tuned | 71.6 | 92.5 | 97.5 | 88.4 | 97.6 | 88.4 | 94.7 | 95.6 | 98.2 | 71.3 |
| **CLIP-ViT-B/16** | | | | | | | | | | |
| Pre-trained | 46.4 | 88.4 | 98.3 | 66.3 | 90.8 | 87.0 | 67.3 | 12.4 | 11.2 | 60.6 |
| Fine-tuned | 72.8 | 94.5 | 98.2 | 88.8 | 98.3 | 91.9 | 94.5 | 95.3 | 98.1 | 75.7 |
| **CLIP-ViT-L/14** | | | | | | | | | | |
| Pre-trained | 50.0 | 93.2 | 99.4 | 75.1 | 95.6 | 91.2 | 67.0 | 12.3 | 9.7 | 68.9 |
| Fine-tuned | 75.9 | 95.7 | 99.2 | 93.0 | 99.1 | 94.8 | 95.3 | 95.4 | 98.3 | 80.5 |

## B.4 Details of Baseline Hyper-parameters

Tab. 11 summarizes the hyper-parameters for all baseline methods under different task configurations (8, 14, 20 tasks). *Top-k* denotes the pruning ratio, *TALL* the TALL mask threshold, and *Cons.* the consensus mask threshold. The column *LR* is the learning rate, while *Steps* indicates the number of adaptation steps. $r$ represents the LoRA rank, and the last column jointly reports the null dimension ($k$), EMA decay ($\beta$), and relaxation coefficient ($\gamma$).

Table 11: hyper-parameter settings for all baselines.

| Method | Tasks | $\lambda$ | Top-k | TALL | Cons. | LR | Steps | $r$ | $k/\beta/\gamma$ |
|---|---|---|---|---|---|---|---|---|---|
| TASK ARITHMETIC | 8 | 0.3 | - | - | - | - | - | - | - |
| | 14/20 | 0.1 | - | - | - | - | - | - | - |
| TIES-MERGING | 8 | 0.3 | 20 | - | - | - | - | - | - |
| | 14/20 | 0.1 | 20 | - | - | - | - | - | - |
| CONSENSUS TA | 8/14/20 | 0.1 | - | 0.2 | 2 | - | - | - | - |
| LW. ADAMERGING | 8/14/20 | 0.3 | - | - | - | 1e-4 | 50 | - | - |
| WEMOE | 8/14/20 | 0.3 | - | - | - | 1e-4 | 50 | - | - |
| LORA-WEMOE | 8/14/20 | 0.3 | - | - | - | 1e-4 | 50 | 64 | - |
| MINGLE-STATIC | 8/14/20 | 0.3 | - | - | - | - | - | - | - |
| **MINGLE (Ours)** | 8/14/20 | - | - | - | - | 1e-4 | 50 | 64 | 3/0.99/1.0 |

## B.5 Comparison of Assumptions and Requirements

Tab. 12 summarizes the assumptions and resource requirements of all baseline methods. We report whether each method requires storing intermediate activations, introduces additional parameters (for storage or inference), and incurs extra test-time computation. Our method only maintains a fixed-size covariance matrix instead of full activations, leading to constant memory regardless of test set size. Although LoRA experts are stored, the router merges them into a single model per input, so the effective inference cost matches that of a standard individual model.

Table 12: Comparison of baseline assumptions and requirements.

| Method | Save Activations | Extra Parameters (Storage) | Extra Parameters (Inference) | Test-time Compute |
|---|---|---|---|---|
| TASK ARITHMETIC | No | No | No | No |
| TIES-MERGING | No | No | No | No |
| MAGMAX-IND | No | No | No | No |
| OPCM | No | No | No | No |
| CONSENSUS TA | No | Yes | No | No |
| LW. ADAMERGING | No | No | No | Yes |
| WEMOE | No | Yes | No | Yes |
| MINGLE-STATIC | No | No | No | No |
| **MINGLE (Ours)** | No[1] | Yes[2] | No[2] | Yes |

[1] Only a fixed-size covariance matrix is maintained, resulting in constant memory regardless of test set size.
[2] LoRA experts are stored, but the router merges them into a single model per input, making the effective inference size equivalent to a standard individual model.

# C  Additional Results

In this section, we provide additional experimental results to support the findings reported in the main paper. Specifically, we include: (1) detailed overall performance results (C.1); (2) accuracy trends across sequential tasks (C.2); (3) detailed results under distribution shifts (C.3); and (4) extended visualizations of gate activations and hyper-parameter effects (C.6).

## C.1  Detailed Overall Performance Results

Tab. 13 expands on the average results in Tab. 1 by reporting per-task average accuracy after continually merging 20 tasks. We compare six methods, SWA, Task Arithmetic, Ties-Merging, MagMax-IND, OPCM, and our proposed MINGLE across three CLIP-ViT backbones (B/32, B/16, L/14). MINGLE achieves the highest accuracy on most tasks. These fine-grained results reinforce the main paper's findings, highlighting MINGLE's ability to improve performance on continual model merging.

## C.2  Accuracy Trends Across Sequential Tasks

Fig. 5 provides a detailed view of accuracy throughout the continual merging process across different settings, showing both the performance on the current task and on previously encountered ones. The progressive accuracy drop across columns illustrates the degree of forgetting over time. Notably, MINGLE consistently alleviates this degradation, demonstrating markedly reduced forgetting across the full task sequence. Fig. 6 further compares the average accuracy curves of MINGLE and baseline methods on previously seen tasks after each new model is merged, using the CLIP ViT-B/16 backbone. Results are averaged over 10 random task orderings. MINGLE consistently achieves the highest performance throughout the merging process, with its accuracy curve clearly dominating those of competing methods. Moreover, the narrower standard deviation bands indicate that MINGLE is more robust to the task orders.

## C.3  Detailed Results Under Distribution Shifts

Tab. 14 expands on Tab. 4 by reporting per-dataset accuracy under both clean test conditions and seven common corruption types: motion blur, impulse noise, Gaussian noise, pixelation, spatter, contrast shift, and JPEG compression. We evaluate six merging methods, across four downstream tasks: Cars, EuroSAT, RESISC45, and GTSRB. This detailed breakdown complements the average results in the main paper, providing a more comprehensive assessment of robustness under test-time distribution shifts.

Table 13: Test set accuracy comparisons on different downstream tasks.

| Model | SUN397 | Cars | RESISC45 | EuroSAT | SVHN | GTSRB | MNIST | DTD | Flowers102 | PCAM |
|---|---|---|---|---|---|---|---|---|---|---|
| **ViT-B/32** | | | | | | | | | | |
| C. Fine-Tuned | 53.9 | 38.2 | 64.7 | 98.7 | 45.4 | 34.4 | 86.7 | 58.4 | 57.5 | 67.7 |
| Average (SWA) | 64.2 | 59.6 | 64.8 | 60.9 | 47.3 | 43.1 | 71.8 | 46.4 | 66.5 | 63.9 |
| C.TA | 62.0 | 53.7 | 60.9 | 58.1 | 48.5 | 48.9 | 79.4 | 46.1 | 61.1 | 73.4 |
| C.TIES | 62.5 | 49.1 | 55.8 | 50.9 | 54.6 | 49.3 | 82.0 | 46.7 | 58.5 | 69.9 |
| MagMax-Ind | 63.6 | 53.1 | 59.7 | 49.1 | 53.8 | 53.1 | 79.8 | 43.2 | 56.9 | 75.1 |
| Consensus TA | 37.0 | 25.2 | 35.2 | 36.7 | 37.3 | 44.1 | 80.6 | 30.3 | 33.5 | 59.2 |
| C. LW AdaMerging | 63.1 | 60.0 | 63.5 | 60.1 | 35.6 | 32.1 | 51.8 | 45.4 | 66.6 | 60.2 |
| C. LoRA-WEMOE | 51.4 | 45.8 | 63.3 | 43.5 | 42.9 | 34.6 | 58.9 | 46.5 | 47.5 | 60.1 |
| OPCM | 64.4 | 51.1 | 66.0 | 71.7 | 66.1 | 56.0 | 90.2 | 40.4 | 64.9 | 80.2 |
| Mingle (Ours) | 67.8 | 58.3 | 83.5 | 90.0 | 82.9 | 91.8 | 98.0 | 65.3 | 74.0 | 66.9 |
| Mingle* (Ours) | 68.8 | 64.2 | 83.8 | 91.1 | 82.4 | 89.0 | 96.9 | 62.8 | 76.7 | 72.8 |
| **ViT-B/16** | | | | | | | | | | |
| C. Fine-Tuned | 62.7 | 58.0 | 67.6 | 99.1 | 46.0 | 29.2 | 93.9 | 61.9 | 64.1 | 75.2 |
| Average (SWA) | 67.1 | 64.6 | 69.3 | 63.4 | 62.4 | 52.7 | 80.7 | 46.6 | 71.8 | 63.1 |
| C.TA | 65.8 | 57.5 | 63.8 | 59.5 | 64.7 | 54.0 | 88.0 | 45.3 | 67.5 | 67.1 |
| C.TIES | 64.2 | 52.9 | 60.9 | 53.0 | 62.8 | 48.8 | 88.4 | 45.0 | 61.3 | 68.5 |
| MagMax-Ind | 65.8 | 51.8 | 57.8 | 42.6 | 54.4 | 43.7 | 83.0 | 42.8 | 60.4 | 69.8 |
| Consensus TA | 42.6 | 24.8 | 30.4 | 34.4 | 47.6 | 42.2 | 79.9 | 30.6 | 36.2 | 74.3 |
| C. LW AdaMerging | 65.5 | 65.7 | 69.8 | 59.4 | 50.1 | 44.2 | 61.1 | 47.1 | 71.8 | 57.9 |
| C. LoRA-WEMOE | 62.7 | 60.2 | 69.4 | 37.7 | 52.1 | 39.9 | 63.1 | 45.3 | 64.3 | 51.7 |
| OPCM | 67.9 | 55.9 | 73.7 | 77.5 | 74.4 | 63.2 | 94.1 | 49.2 | 72.3 | 79.6 |
| Mingle (Ours) | 71.5 | 64.9 | 85.3 | 90.0 | 87.5 | 90.1 | 97.1 | 62.7 | 82.6 | 80.6 |
| Mingle* (Ours) | 72.0 | 72.1 | 87.9 | 93.3 | 87.1 | 89.2 | 97.4 | 62.5 | 86.8 | 76.4 |
| **ViT-L/14** | | | | | | | | | | |
| C. Fine-Tuned | 69.5 | 73.6 | 78.3 | 99.2 | 59.3 | 49.3 | 98.6 | 69.7 | 83.2 | 78.3 |
| Average (SWA) | 70.7 | 77.7 | 76.4 | 75.3 | 69.5 | 62.1 | 93.7 | 57.7 | 80.0 | 73.6 |
| C.TA | 70.4 | 74.1 | 73.9 | 66.3 | 69.9 | 65.6 | 95.1 | 56.6 | 78.6 | 70.4 |
| C.TIES | 69.7 | 70.3 | 65.3 | 47.9 | 76.1 | 63.6 | 94.7 | 54.4 | 77.9 | 72.3 |
| MagMax-Ind | 73.1 | 73.7 | 75.6 | 64.6 | 73.7 | 68.8 | 94.6 | 56.1 | 78.0 | 71.7 |
| Consensus TA | 50.7 | 39.1 | 31.7 | 36.4 | 39.4 | 44.9 | 88.5 | 33.8 | 45.7 | 62.5 |
| C. LW AdaMerging | 68.8 | 78.6 | 75.9 | 65.7 | 58.3 | 51.6 | 79.9 | 57.4 | 80.6 | 52.4 |
| C. LoRA-WEMOE | 62.1 | 68.1 | 68.7 | 53.2 | 47.5 | 49.4 | 69.8 | 49.1 | 66.2 | 54.2 |
| OPCM | 73.1 | 78.3 | 82.4 | 80.2 | 80.8 | 80.4 | 97.4 | 61.6 | 84.8 | 76.3 |
| Mingle (Ours) | 75.9 | 83.4 | 87.8 | 88.7 | 91.1 | 94.5 | 98.4 | 70.8 | 94.8 | 75.3 |
| Mingle* (Ours) | 74.5 | 85.9 | 90.5 | 92.5 | 90.1 | 92.7 | 98.1 | 69.2 | 95.7 | 74.0 |

| Model | FER2013 | OxfordIIITPet | STL10 | CIFAR100 | CIFAR10 | Food101 | FashionMNIST | EMNIST | KMNIST | RenderedSST2 |
|---|---|---|---|---|---|---|---|---|---|---|
| **ViT-B/32** | | | | | | | | | | |
| C. Fine-Tuned | 58.3 | 68.5 | 86.7 | 40.2 | 70.5 | 50.0 | 90.7 | 72.4 | 54.5 | 54.5 |
| Average (SWA) | 50.2 | 84.1 | 97.0 | 69.8 | 92.7 | 80.4 | 71.3 | 15.0 | 11.5 | 61.8 |
| C.TA | 51.4 | 82.3 | 94.9 | 64.6 | 91.4 | 71.9 | 73.9 | 17.8 | 12.2 | 59.9 |
| C.TIES | 49.5 | 81.3 | 95.2 | 63.7 | 91.2 | 70.2 | 73.7 | 17.8 | 16.9 | 59.8 |
| MagMax-Ind | 56.5 | 79.9 | 94.6 | 58.7 | 91.9 | 73.8 | 74.3 | 18.3 | 15.4 | 63.9 |
| Consensus TA | 41.7 | 58.8 | 81.8 | 41.5 | 78.1 | 29.8 | 72.6 | 17.4 | 18.5 | 54.1 |
| C. LW AdaMerging | 43.2 | 83.7 | 96.8 | 67.0 | 89.9 | 81.6 | 63.7 | 16.8 | 10.7 | 59.1 |
| C. LoRA-WEMOE | 44.6 | 72.5 | 86.1 | 40.1 | 63.8 | 63.8 | 48.1 | 10.3 | 12.8 | 55.7 |
| OPCM | 58.5 | 82.9 | 95.9 | 67.6 | 92.8 | 74.0 | 76.3 | 22.4 | 18.3 | 64.6 |
| Mingle (Ours) | 65.0 | 85.5 | 97.0 | 72.6 | 94.1 | 81.5 | 85.4 | 50.4 | 65.2 | 67.1 |
| Mingle* (Ours) | 65.3 | 88.5 | 97.7 | 73.9 | 94.7 | 83.7 | 86.4 | 39.3 | 56.1 | 68.7 |
| **ViT-B/16** | | | | | | | | | | |
| C. Fine-Tuned | 60.5 | 84.5 | 90.5 | 38.8 | 73.6 | 61.9 | 89.7 | 83.3 | 51.5 | 72.8 |
| Average (SWA) | 50.9 | 89.6 | 98.0 | 72.9 | 94.2 | 85.9 | 73.3 | 15.6 | 12.4 | 62.5 |
| C.TA | 50.7 | 89.3 | 97.0 | 68.0 | 93.1 | 80.3 | 75.7 | 18.1 | 16.7 | 61.8 |
| C.TIES | 50.4 | 87.9 | 96.3 | 63.1 | 91.7 | 78.0 | 75.0 | 23.4 | 24.9 | 61.5 |
| MagMax-Ind | 57.7 | 88.8 | 97.5 | 71.5 | 94.4 | 81.3 | 77.2 | 24.5 | 25.0 | 59.4 |
| Consensus TA | 45.6 | 76.8 | 87.7 | 44.4 | 82.2 | 38.4 | 72.7 | 18.8 | 30.0 | 58.6 |
| C. LW AdaMerging | 46.8 | 88.9 | 98.1 | 69.2 | 91.4 | 86.6 | 67.2 | 17.2 | 11.0 | 59.2 |
| C. LoRA-WEMOE | 45.6 | 91.2 | 92.3 | 41.3 | 64.3 | 78.1 | 48.0 | 23.5 | 16.6 | 52.7 |
| OPCM | 59.5 | 91.8 | 97.7 | 73.2 | 94.7 | 83.1 | 81.3 | 26.5 | 23.4 | 66.8 |
| Mingle (Ours) | 67.6 | 92.7 | 97.4 | 74.0 | 95.3 | 87.7 | 87.4 | 73.5 | 79.9 | 74.0 |
| Mingle* (Ours) | 67.9 | 93.5 | 98.4 | 77.7 | 96.4 | 89.7 | 87.8 | 56.6 | 64.5 | 75.3 |
| **ViT-L/14** | | | | | | | | | | |
| C. Fine-Tuned | 68.0 | 92.1 | 94.5 | 60.5 | 85.7 | 74.8 | 93.1 | 89.0 | 59.2 | 78.8 |
| Average (SWA) | 52.7 | 94.2 | 99.2 | 81.7 | 97.0 | 90.7 | 77.4 | 16.1 | 10.4 | 66.1 |
| C.TA | 55.7 | 94.2 | 98.6 | 79.1 | 96.6 | 87.6 | 80.8 | 17.6 | 10.6 | 63.6 |
| C.TIES | 57.6 | 93.5 | 97.8 | 74.0 | 95.6 | 84.7 | 79.7 | 20.2 | 12.6 | 58.4 |
| MagMax-Ind | 52.9 | 93.9 | 98.7 | 82.1 | 97.3 | 89.5 | 81.6 | 19.2 | 11.1 | 68.4 |
| Consensus TA | 50.3 | 82.2 | 89.7 | 47.5 | 86.2 | 43.5 | 75.3 | 14.5 | 10.4 | 53.4 |
| C. LW AdaMerging | 49.2 | 93.5 | 99.3 | 77.2 | 95.8 | 91.1 | 68.2 | 18.6 | 9.8 | 66.6 |
| C. LoRA-WEMOE | 46.3 | 84.5 | 87.6 | 52.1 | 70.5 | 73.3 | 50.0 | 18.7 | 10.9 | 56.5 |
| OPCM | 61.8 | 95.4 | 99.2 | 83.0 | 97.8 | 90.9 | 86.0 | 26.4 | 14.7 | 71.0 |
| Mingle (Ours) | 67.7 | 96.0 | 98.7 | 81.4 | 97.1 | 90.6 | 90.6 | 60.7 | 88.6 | 79.8 |
| Mingle* (Ours) | 67.9 | 96.0 | 99.4 | 84.7 | 97.8 | 92.4 | 88.8 | 53.0 | 57.1 | 75.5 |

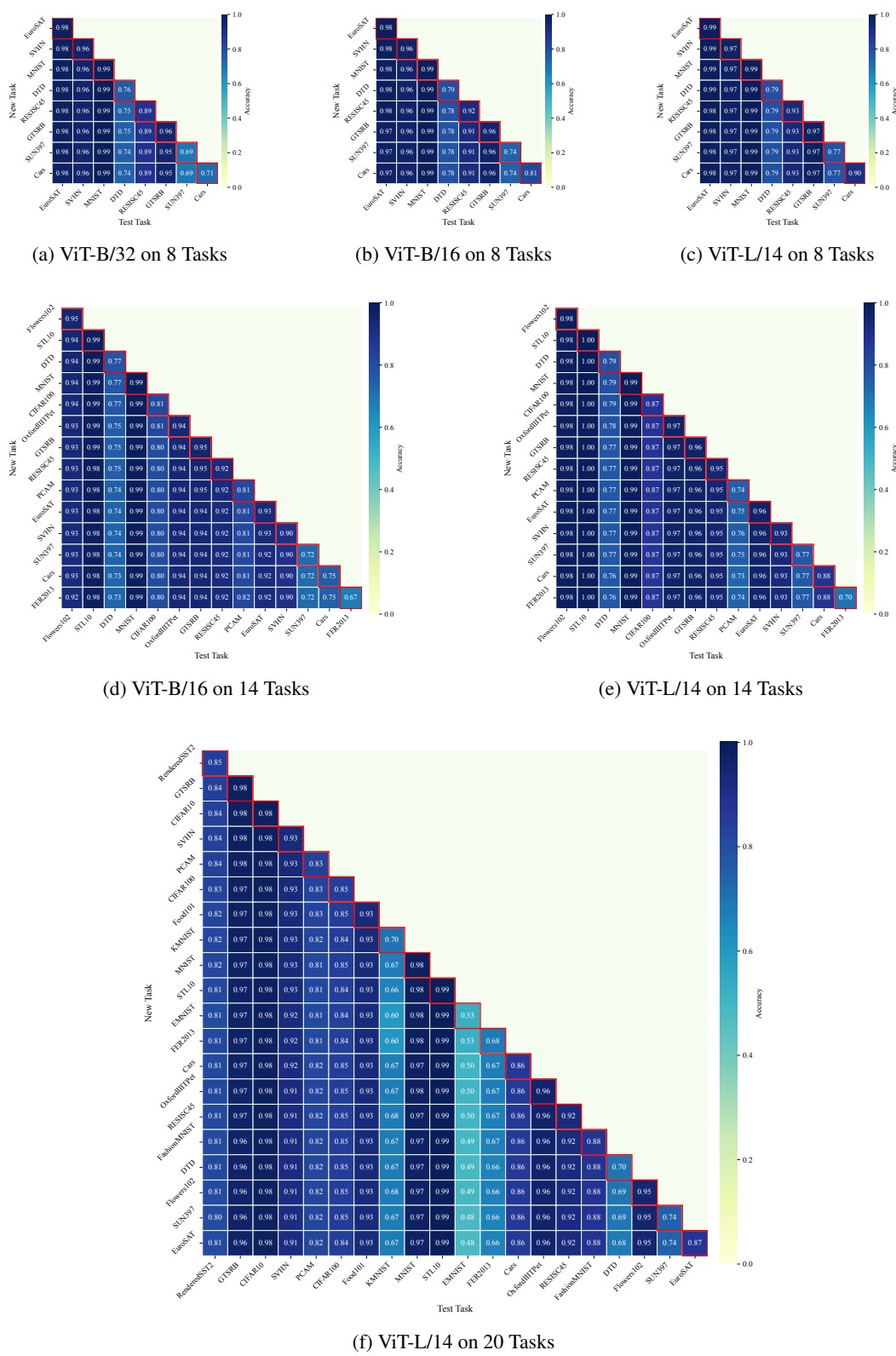

Figure 5: Accuracy matrices of MINGLE (ViT-B/32, ViT-B/16, and ViT-L/14) under different task settings.

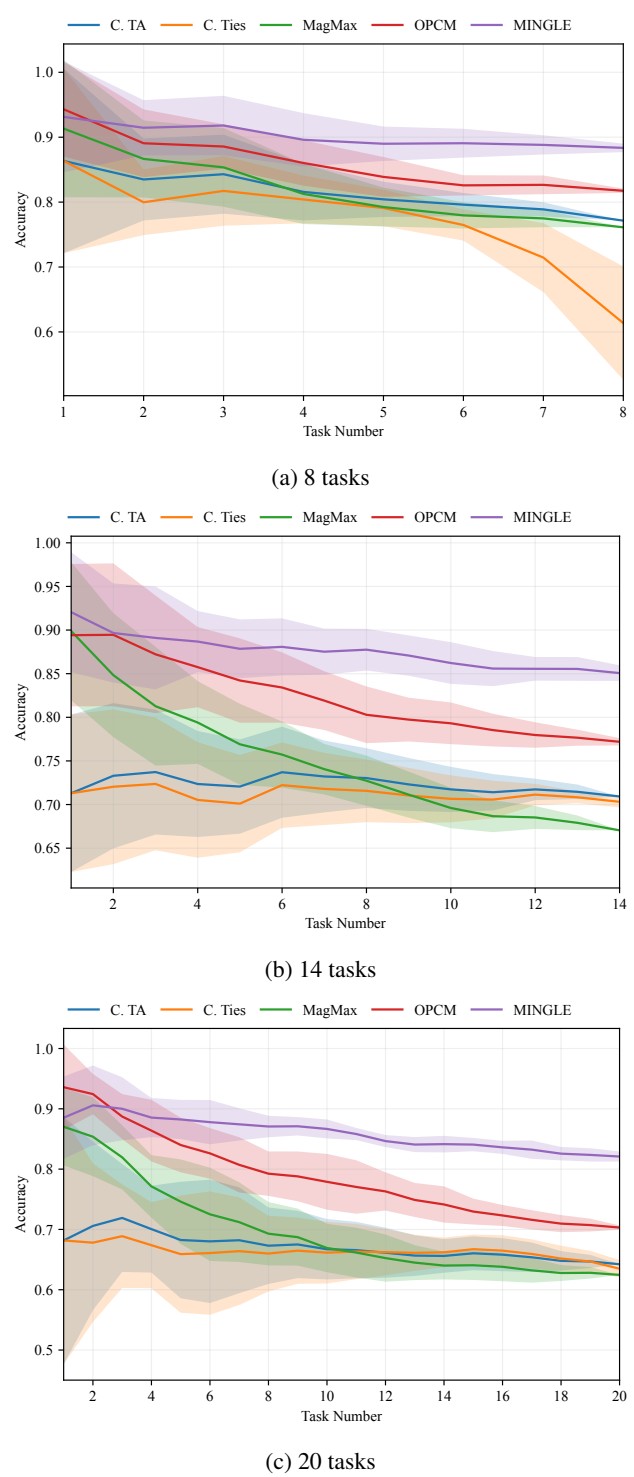

(a) 8 tasks

(b) 14 tasks

(c) 20 tasks

Figure 6: Sequential test accuracy curves of MINGLE and baselines (C.TA, C.Ties, MagMax, OPCM) under different task settings. Shaded regions indicate standard deviation across 10 task orders.

Table 14: Robustness results when merging ViT-B/32 models on four tasks.

| Method | Cars | EuroSAT | RESISC45 | GTSRB | Avg ACC | Cars | EuroSAT | RESISC45 | GTSRB | Avg ACC |
|---|---|---|---|---|---|---|---|---|---|---|
| | **Clean Test Set** | | | | | **Corruption: Motion Blur** | | | | |
| C. LW ADAMERGING | 65.3 | 49.7 | 65.4 | 43.8 | 56.0 | 64.2 | 25.6 | 62.5 | 37.5 | 47.5 |
| C. WEMOE | 0.5 | 8.1 | 2.6 | 2.5 | 3.4 | 0.5 | 8.0 | 1.8 | 2.3 | 3.1 |
| C. LORA-WEMOE | 66.1 | 84.3 | 81.0 | 83.6 | 78.7 | 64.8 | 57.9 | 82.0 | 79.2 | 71.0 |
| C. TASK ARITHMETIC | 64.6 | 90.4 | 80.2 | 74.8 | 77.5 | 62.3 | 59.4 | 78.5 | 63.3 | 65.9 |
| MAGMAX-IND | 63.1 | 89.2 | 81.7 | 82.5 | 79.1 | 61.4 | 62.1 | 80.0 | 72.6 | 69.0 |
| OPCM | 65.7 | 92.3 | 85.7 | 90.5 | 83.6 | 62.8 | 62.5 | 83.7 | 82.2 | 72.8 |
| MINGLE (Ours) | 74.4 | 96.5 | 91.5 | 97.3 | 89.9 | 73.2 | 70.5 | 91.9 | 95.8 | 82.9 |
| | **Corruption: Impulse Noise** | | | | | **Corruption: Gaussian Noise** | | | | |
| C. LW ADAMERGING | 60.5 | 30.1 | 56.3 | 25.5 | 43.1 | 62.3 | 25.6 | 59.7 | 25.6 | 43.3 |
| C. WEMOE | 0.5 | 11.2 | 2.3 | 3.2 | 4.3 | 0.5 | 8.1 | 2.4 | 2.8 | 3.4 |
| C. LORA-WEMOE | 62.2 | 23.4 | 69.9 | 64.6 | 55.0 | 64.9 | 31.7 | 77.8 | 63.4 | 59.4 |
| C. TASK ARITHMETIC | 59.9 | 57.7 | 72.9 | 45.0 | 58.9 | 61.8 | 51.4 | 75.1 | 50.1 | 59.6 |
| MAGMAX-IND | 59.2 | 56.3 | 74.3 | 52.5 | 60.6 | 60.6 | 51.7 | 77.0 | 56.5 | 61.5 |
| OPCM | 61.1 | 57.1 | 78.5 | 62.0 | 64.7 | 63.0 | 52.4 | 80.7 | 64.9 | 65.2 |
| MINGLE (Ours) | 69.6 | 28.0 | 86.1 | 86.1 | 67.5 | 72.0 | 38.5 | 89.4 | 82.9 | 70.7 |
| | **Corruption: Pixelate** | | | | | **Corruption: Spatter** | | | | |
| C. LW ADAMERGING | 3.4 | 16.5 | 13.5 | 39.2 | 18.1 | 61.3 | 34.1 | 58.2 | 32.8 | 46.6 |
| C. WEMOE | 0.5 | 6.3 | 2.5 | 2.5 | 3.0 | 0.5 | 10.1 | 2.7 | 2.6 | 4.0 |
| C. LORA-WEMOE | 0.8 | 26.0 | 5.8 | 67.0 | 24.9 | 62.4 | 35.4 | 71.2 | 73.0 | 60.5 |
| C. TASK ARITHMETIC | 2.5 | 31.7 | 19.1 | 65.6 | 29.7 | 61.2 | 63.1 | 72.7 | 57.0 | 63.5 |
| MAGMAX-IND | 2.6 | 36.1 | 19.3 | 74.0 | 33.0 | 60.0 | 64.9 | 74.8 | 66.1 | 66.4 |
| OPCM | 2.1 | 34.3 | 19.5 | 84.9 | 35.2 | 61.5 | 64.7 | 78.8 | 76.8 | 70.5 |
| MINGLE (Ours) | 2.3 | 35.6 | 18.5 | 95.1 | 37.9 | 70.1 | 57.8 | 86.2 | 93.9 | 77.0 |
| | **Corruption: Contrast** | | | | | **Corruption: JPEG Compression** | | | | |
| C. LW ADAMERGING | 61.8 | 26.0 | 63.1 | 44.8 | 48.9 | 65.1 | 29.6 | 65.4 | 36.4 | 49.1 |
| C. WEMOE | 0.5 | 7.5 | 2.3 | 3.0 | 3.3 | 0.5 | 10.5 | 2.4 | 2.7 | 4.0 |
| C. LORA-WEMOE | 64.3 | 46.5 | 77.7 | 85.6 | 68.5 | 65.5 | 59.1 | 80.4 | 74.0 | 69.7 |
| C. TASK ARITHMETIC | 62.5 | 55.2 | 75.3 | 70.8 | 66.0 | 64.1 | 66.2 | 80.0 | 61.0 | 67.8 |
| MAGMAX-IND | 61.3 | 58.0 | 76.9 | 78.2 | 68.6 | 62.5 | 67.7 | 81.1 | 68.5 | 69.9 |
| OPCM | 63.8 | 57.5 | 81.3 | 87.4 | 72.5 | 65.0 | 68.0 | 85.4 | 79.3 | 74.4 |
| MINGLE (Ours) | 72.4 | 60.1 | 90.4 | 97.3 | 80.1 | 73.7 | 73.5 | 92.0 | 92.4 | 82.9 |

## C.4 Inference Efficiency and Parameter Overhead

Tab. 15 compares the inference efficiency and parameter overhead of all baselines on the CLIP ViT-B/32 model after merging eight tasks. We report the total number of parameters, additional storage and inference parameters, throughput (images per second), and accuracy. The results show that most static merging methods (*e.g.*, Task Arithmetic, Ties-Merging, MAGMAX-Ind, OPCM) incur no extra storage or inference overhead, but typically achieve limited accuracy. Consensus TA and WEMOE introduce significant storage overhead, while WEMOE also scales up inference parameters considerably. By contrast, MINGLE achieves a favorable trade-off: although it introduces additional parameters for LoRA experts and the router, the effective inference overhead remains small, and throughput is only marginally reduced compared to static baselines. This efficiency advantage comes while delivering substantially higher accuracy.

## C.5 Forward Transfer Analysis

Forward transfer (FWT) is an important metric in continual learning, as it quantifies how effectively prior knowledge facilitates the learning of future tasks. We adopt the standard definition:

$$\text{FWT} = \frac{1}{T-1} \sum_{t=2}^{T} \left[ a_t(\theta_t^{\text{merged}}) - \bar{a}_t \right], \tag{30}$$

where $a_t(\theta_t^{\text{merged}})$ denotes the test accuracy on task $t$ using the merged model after task $t$, and $\bar{a}_t$ is the accuracy of the individually fine-tuned model for task $t$. Positive FWT indicates beneficial transfer, while negative values suggest interference.

Table 15: Comparison of inference efficiency and parameter overhead on CLIP ViT-B/32 model after eight tasks merging.

| Method | Total Params (M) | Extra Storage (M) | Extra Inference (M) | Throughput (img/s) | ACC (%) |
|---|---|---|---|---|---|
| TASK ARITHMETIC | 87.5 | 0.0 | 0.0 | ∼910 | 67.5 |
| TIES-MERGING | 87.5 | 0.0 | 0.0 | ∼910 | 49.0 |
| MAGMAX-IND | 87.5 | 0.0 | 0.0 | ∼910 | 70.7 |
| OPCM | 87.5 | 0.0 | 0.0 | ∼910 | 75.5 |
| CONSENSUS TA | 87.5 | 87.5 | 0.0 | ∼910 | 69.0 |
| LW. ADAMERGING | 87.5 | 0.0 | 0.0 | ∼910 | 52.9 |
| WEMOE | 540.9 | 453.4 | 0.07 | ∼858 | 4.9 |
| WEMOE-LoRA | 103.7 | 16.2 | 0.07 | ∼848 | 66.6 |
| MINGLE | 173.1 | 85.6 | 0.6 | ∼841 | 85.8 |
| MINGLE* | 113.7 | 26.2 | 0.3 | ∼862 | 85.0 |

Table 16: Forward transfer (FWT) results on 8-task continual merging with CLIP ViT-B/16.

| Method | ACC (%) | BWT (%) | FWT (%) |
|---|---|---|---|
| TASK ARITHMETIC | 77.1 ± 0.0 | -4.2 ± 1.0 | -13.4 ± 0.0 |
| TIES-MERGING | 66.8 ± 3.7 | -5.5 ± 0.4 | -30.7 ± 9.9 |
| OPCM | 81.8 ± 0.3 | -4.8 ± 0.7 | -9.0 ± 0.4 |
| **MINGLE (Ours)** | 88.3 ± 0.6 | -0.4 ± 0.1 | -3.8 ± 0.8 |

Tab. 16 reports the results for the 8-task continual merging setup on CLIP ViT-B/16. The results demonstrate that MINGLE achieves nearly zero forgetting (BWT ≈ 0) while obtaining the highest forward transfer among all baselines, showing that our adaptive gating and merging strategy not only preserves past knowledge but also enhances feature utility for future tasks.

## C.6 Additional Visualizations of Gate Activations and the Relaxation Effect

We provide an extended ablation study on gate hyper-parameters, including visualizations of gate activations under 14-task (Fig. 7) and 20-task (Fig. 8) configurations, complementing the 8-task results presented in the main paper. The visualizations demonstrate that the null-space constraint remains effective as the number of tasks increases, consistently suppressing gate responses to inputs from previously seen tasks and thereby mitigating forgetting.

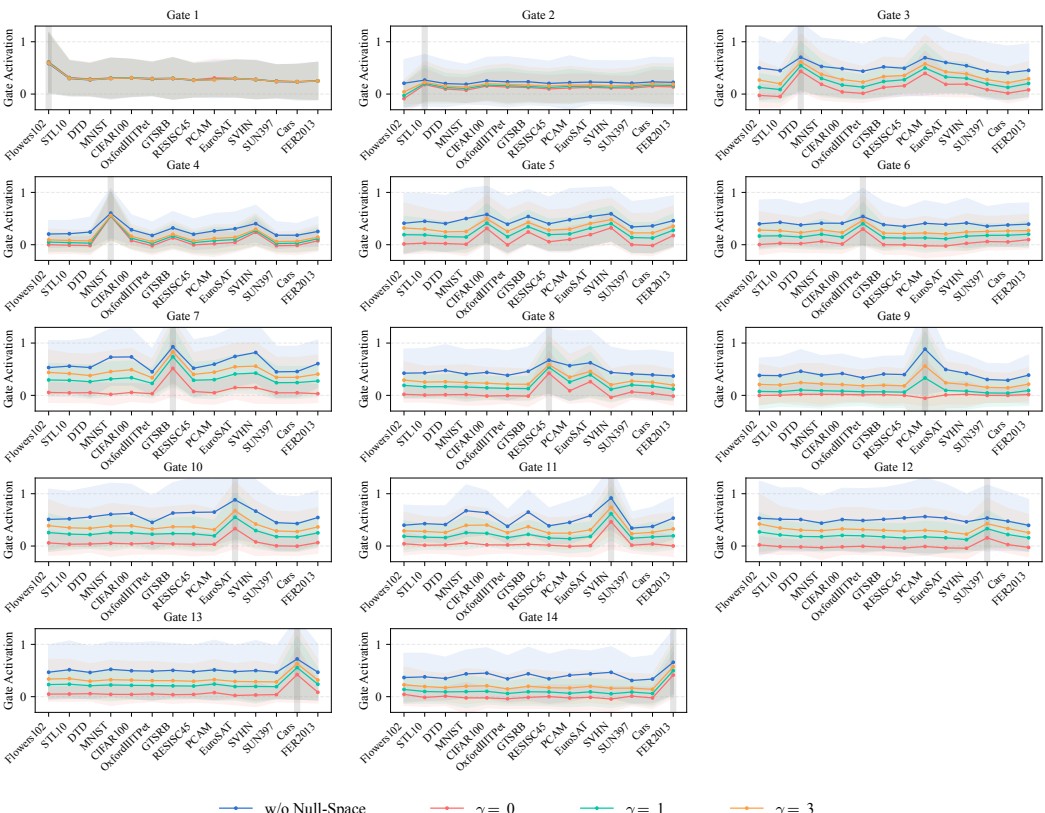

Figure 7: Visualization of gate activations across 14 tasks under varying $\gamma$ values. Each subplot corresponds to a gate, with curves and shaded regions denoting the mean and standard deviation of activations across layers. Gray bars mark the training dataset for each gate. Smaller $\gamma$ values result in stronger suppression of activations on previously learned tasks.

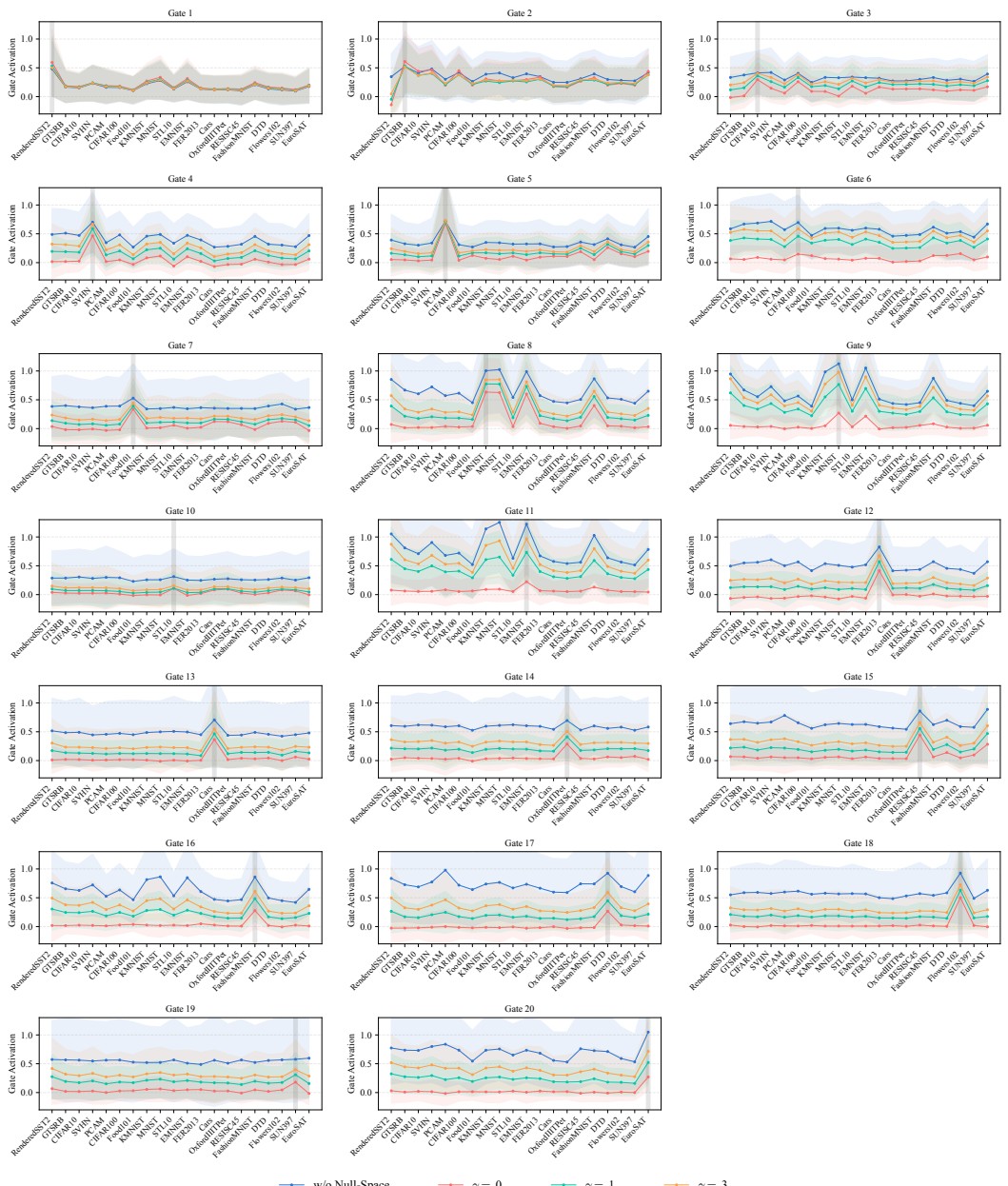

Figure 8: Visualization of gate activations across 20 tasks under varying $\gamma$ values.

# D Discussions

## D.1 Use of Unlabeled Adaptation Samples

In our experiments, we simulate a realistic deployment setting by randomly sampling 5 unlabeled examples per class from the test split, which serve as adaptation samples for model merging. Such small unlabeled buffers are practical in real-world applications and can be obtained from various sources, including (i) incoming test-time data such as recent user queries or model inputs, (ii) held-out validation inputs or small training subsets (if available), (iii) user-provided samples (*e.g.*, few-shot examples) that do not raise privacy concerns, (iv) synthetically generated data, or (v) manually curated public data. Importantly, our method does not depend on precise sample selection, and the buffer size remains fixed and small, ensuring feasibility and robustness in deployment scenarios.

## D.2 Relation to Rehearsal-Free Continual Learning

Test-time continual model merging (TTCMM) is closely related to the paradigm of rehearsal-free continual learning (RFCL), as both approaches share two fundamental constraints: (i) they do not retain past training data, and (ii) they avoid storing previous task models. The key distinction lies in the information available at each stage. RFCL assumes access to the training data of the current task and incrementally fine-tunes a single model over time. In contrast, TTCMM assumes access to independently fine-tuned models for each new task and focuses on merging these expert models rather than training them from scratch. Additionally, TTCMM relies on a small unlabeled buffer at test time (*e.g.*, 5 samples per class) to guide the merging process.

From a privacy perspective, TTCMM provides stronger guarantees. Since it does not require access to full training sets, it only depends on a small set of unlabeled samples, which can be user-provided without risk, synthetically generated, or curated from public data. By comparison, RFCL requires access to large-scale labeled datasets for every task, raising more significant concerns regarding privacy, storage, and legal constraints (*e.g.*, medical images, personal data, or copyrighted corpora). The reliance on a tiny unlabeled buffer makes TTCMM more practical in scenarios where data privacy is a primary consideration.

## D.3 Limitations

As with many model merging methods, our approach assumes that all independently fine-tuned models originate from a shared pretrained initialization. The extent to which this assumption influences merging performance remains unclear and warrants further investigation. In addition, our current experiments focus on merging models with identical backbone architectures (*e.g.*, CLIP ViT-B/16). Although our use of LoRA-based expert offers some structural uniformity, which could potentially accommodate heterogeneous backbones, we have not yet explored this setting. Extending our framework to support diverse initialization points or architectural variants remains an open direction for future work.

## D.4 Broader Impacts

This paper presents work whose goal is to advance the field of Machine Learning. There are many potential societal consequences of our work, none which we feel must be specifically highlighted here.

