# OpenReview forum: "MINGLE: Mixture of Null-Space Gated Low-Rank Experts for Test-Time Continual Model Merging"
_NeurIPS.cc/2025/Conference — NeurIPS 2025 poster_

### Official Review · Reviewer_np8P · 2025-06-12

**Clarity:** 2
**Significance:** 2
**Originality:** 3
**Rating:** 3
**Confidence:** 3

**Summary:**

This paper proposes mingle for test-time continual model merging, which alleviates the conflict between tasks through mixtures of lora experts and adaptive null-space constrained gating, bringing significant improvement compared to the existing continual model merging methods.

**Questions:**

+ Can you provide a comparative experiment on efficiency with baselines?
+ Is there any forward transfer?

**Ethical Concerns:**

["NO or VERY MINOR ethics concerns only"]

**Final Justification:**

I maintain my score for the following reasons:
+ I agree that the proposed scenario test-time continual model merging is reasonable and meaningful to some extent.
+ However, the assumption of the proposed scenario is too strong and the application scenario is limited. The proposed test-time continual model merging must assume a scenario like some pre-trained models are released by someone and then must be deleted. It is reasonable to delete data due to privacy, but it is kind of strange to delete a model to be merged.
+ From my point of view, test-time adaptation is to use some current unlabeled test samples to adapt the model, thus the model can always be adapted if you need to test or use it. Thus **the authors do not adapt the model for task t-1 test samples after merging task t** is kind of strange as well, especially when the experts of previous tasks are preserved.

**Limitations:**

+ Extending their framework to support diverse initialization points or architectural variants remains an open direction for future work.

**Paper Formatting Concerns:**

No line space after the table title.

**Quality:**

3

**Strengths And Weaknesses:**

+ Strength

    + Through a lightweight seed dataset, the model merging effect is significantly improved with slightly increased computational overhead.
    + Clearly illustrate the advantages of the moe structure over traditional continual model mergin.

+ Weakness

    +  Lack of comparative experiments with baselines on efficiency, storage overhead.
    + The relationship between continual model merging and rehearsal-free continual learning needs to be discussed. Because both continual model merging and rehearsal-free continual learning do not have training data of previous tasks when learning task t, both require training data of the current task and need to have learned models of previous tasks. I do not understand why continual model merging brings about improved privacy.
    + If I understand correctly, the advantage of continual model merging may be that it can train models for t tasks separately and solve any permutation and combination of these t tasks, which is the flexibility. The models for each task must be saved to achieve this flexibility. Given conditions above, if test-time adaptation is added to continual model merging, I have the following concerns: doing **independent** test-time adaptation for **the downstream task currently being tested** may achieve better results because there is no need to consider conflicts between tasks.

---

> ### Author Rebuttal · Authors · 2025-07-31
>
> # **W1 – Comparison of Efficiency and Storage Overhead with Baselines**
>
> ### **Table Z.1: Comparison of inference efficiency and parameter overhead on CLIP ViT-B/32 model after eight tasks merging**
>
> | Method          | Total Params (M) | Extra Storage Params (M) | Extra Inference Params (M) | Throughput (img/s) | ACC (%) |
> | --------------- | ---------------- | ------------------------ | -------------------------- | ------------------ | ------- |
> | Task Arithmetic | 87.5             | 0.0                      | 0.0                        | ~910               | 67.5    |
> | Ties-Merging    | 87.5             | 0.0                      | 0.0                        | ~910               | 49.0    |
> | MAGMAX-Ind      | 87.5             | 0.0                      | 0.0                        | ~910               | 70.7    |
> | OPCM            | 87.5             | 0.0                      | 0.0                        | ~910               | 75.5    |
> | Consensus TA    | 87.5             | 87.5                     | 0.0                        | ~910               | 69.0    |
> | LW. AdaMerging  | 87.5             | 0.0                      | 0.0                        | ~910               | 52.9    |
> | WEMOE           | 540.9            | 453.4                    | 0.07                       | ~858               | 4.9     |
> | WEMOE-LoRA      | 103.7            | 16.2                     | 0.07                       | ~848               | 66.6    |
> | MINGLE          | 173.1            | 85.6                     | 0.6                        | ~841               | 85.8    |
> | MINGLE*         | 113.7            | 26.2                     | 0.3                        | ~862               | 85.0    |
>
>
>
> We sincerely thank the reviewer for raising this point—efficiency and overhead are indeed critical factors for real-world deployments, and we greatly appreciate the suggestion to expand this comparison.
>
> In response, we have included a consolidated table (Table Z.1) comparing **inference speed** and **storage overhead** across all baselines, which demonstrates the practical efficiency of **MINGLE**. Despite employing MoE-style routing, MINGLE incurs **minimal additional inference overhead**, with only a small number of extra parameters used for the gating mechanism (i.e., the MoE routing gates).  The **inference throughput is comparable to static merging methods**, only slightly lower due to the additional MoE parameters, **while providing substantial accuracy improvements**.
>
>
>
>
> # **W2 – Discussion to Rehearsal-Free Continual Learning (RFCL)**
>
> We thank the reviewer for the insightful question, and for prompting a more precise comparison between TTCMM and **rehearsal-free continual learning (RFCL)**—a widely studied paradigm.
>
> Indeed, TTCMM is inspired by RFCL and shares important constraints:
>
> * both **do not retain past training data**, and
> * both **avoid storing previous task models**.
>
> However, the **core difference lies in what is received at each stage**:
>
> * **RFCL** assumes access to the **training data** of the current task, and incrementally fine-tunes a single model over time.
> * **TTCMM**, by contrast, assumes access to an **already independently fine-tuned model** for each new task—thus focusing on **merging** existing expert models rather than learning each from scratch. Additionally, TTCMM requires a **small unlabeled test-time buffer** (e.g., 5 samples per class) to guide the merge.
>
> **On privacy benefits:**
> While both RFCL and TTCMM operate without historical data, TTCMM offers improved privacy guarantees for the following reasons:
>
> 1. **TTCMM does not require access to full training sets.** Instead, it only uses a **tiny** buffer (e.g., 5 **unlabeled** samples per class) from the **test or calibration set**.
> 2. These small unlabeled samples can be:
> *  incoming test-time data (e.g., recent user queries or model inputs),
> *  held-out validation inputs or small training subsets (if available),
> *  user-provided samples without privacy risks (e.g., few-shot examples),
> *  synthetically generated,
> *  or manually curated from public data.
>
> In contrast, RFCL relies on **large-scale labeled** training data for each task, which raises significant privacy, storage, and legal concerns (e.g., medical images, user data, copyrighted materials). In comparison, small unlabeled samples are easier to obtain while maintaining privacy protection.
> # **W3 – On Flexibility and Test-Time Adaptation in TTCMM**
> We sincerely thank the reviewer for their insightful reflection. Your interpretation of TTCMM as offering high flexibility through the ability to merge independently trained task-specific models in arbitrary orders is absolutely correct—and we deeply appreciate your articulation of this strength.
>
> You also raise an important point: that *independently applying TTA to each downstream task* might yield better performance in some cases, as it avoids reconciling conflicts across tasks. We agree that this is indeed a powerful strategy—but only under the assumption that **all task-specific models are available at inference time**.
>
> However, we respectfully emphasize that this assumption does not hold in the **TTCMM setting**, which imposes strict continual learning constraints:
>
> -  **Previous expert models are not retained**—once merged, earlier models are discarded.
> -  The protocol is **strictly incremental**, with one expert model arriving at a time; merging must maintain knowledge across tasks *without* re-accessing past models or data.
>
> Thus, the setting envisioned in your comment—while valid and practically useful—**requires simultaneous access to all expert models**, and is closer in nature to **multi-task model merging  with task-specific adaptation**, which lies **outside the scope of TTCMM**.
>
> We fully agree that this contrast is important and worth clarifying more explicitly. In the revision, we will revise Sec. 1 and Sec. 3.2 to:
>
> * More clearly distinguish TTCMM from settings that assume full model access.
> * Emphasize that TTCMM’s flexibility stems from supporting *arbitrary task sequences*, **but only when models are provided incrementally**, not jointly.
>
> # **Q2 – On Forward Transfer**
>
> We sincerely thank the reviewer for raising this insightful question. Forward transfer is indeed a critical capability in continual learning, measuring how well prior knowledge aids the learning of future tasks. Your suggestion meaningfully strengthens the completeness of our evaluation.
>
> To quantify forward transfer (FWT), we adopt the standard definition:
> $$
> \text{FWT} = \frac{1}{T-1} \sum_{t=2}^{T} \left[ a_t(\theta_t^{merged}) - \bar{a}_t \right]
> $$
>
> where $a_t(\theta_t^{merged})$ denotes the test accuracy on task $t$ using the  model merged after task $t$, and $\bar{a}_t$ is the test accuracy of task $t$ individual finetuned model. Positive FWT indicates beneficial transfer, while negative values suggest interference.
>
> We report the following results for the 8-task continual merging setup with CLIP ViT-B/16:
>
> | Method            | ACC (%)     | BWT (%)     | FWT (%)     |
> | ----------------- | ----------- | ----------- | ----------- |
> | Task Arithmetic   | 77.1 ± 0.0  | -4.2 ± 1.0  | -13.4 ± 0.0 |
> | Ties-Merging      | 66.8 ±  3.7 | -5.5 ± 0.4  | -30.7 ± 9.9 |
> | OPCM              | 81.8  ± 0.3 | -4.8  ± 0.7 | -9.0 ± 0.4  |
> | **MINGLE (Ours)** | 88.3 ± 0.6  | -0.4 ± 0.1  | -3.8 ± 0.8  |
>
> These results show that **MINGLE achieves both zero forgetting (BWT ≈ 0) and the highest forward transfer**, indicating that our adaptive gating and merging pipeline not only preserves knowledge but also improves representation utility for downstream tasks.
>
> We will include these results and the FWT definition in the final version for completeness.
>
> We sincerely thank the reviewer again, and hope our clarifications will support a higher evaluation.

---

> > ### Comment · Reviewer_np8P · 2025-08-04
> >
> > ## Follow-up of W2 & W3
> > + As shown in the rebuttal W3, the author said "TTCMM as offering high flexibility through the ability to merge independently trained task-specific models in arbitrary orders is absolutely correct".
> > + However, as shown in the rebuttal W3, the author said "Previous expert models are not retained—once merged, earlier models are discarded."
> > + This means when the merging begins, the method can merge independently trained task-specific models in only one order.
> > + Therefore, I think some content in the rebuttal is inconsistent. Could the author please explain why the flexibility still exists if previous experts were discarded?

---

> > > ### Author Response · Authors · 2025-08-04
> > >
> > > We thank the reviewer for the follow-up and sincerely apologize for the lack of clarity in our previous rebuttal.
> > >
> > > To clarify: from the **model merging algorithm’s** perspective, TTCMM does **not** support arbitrary merging orders once the process begins—previous expert models are discarded, and merging is performed incrementally using only the current merged model and the newly arriving expert.
> > >
> > > The **flexibility we refer to lies on the model provider's side**, not the merger's. Specifically:
> > >
> > > * In traditional continual learning, task models are typically trained in a fixed sequence, where later models depend on earlier ones.
> > > * In contrast, **TTCMM allows each expert model to be trained independently and delivered in any order**.
> > >
> > > This decoupling offers practical flexibility: task-specific models can be developed and transmitted **in arbitrary sequences**, depending on real-world constraints (e.g., task readiness, institutional availability), and **TTCMM will still operate correctly under such conditions**. Once a sequence is selected and merging begins, the order becomes fixed and earlier models are no longer retained.

---

> > > > ### Comment · Reviewer_np8P · 2025-08-04
> > > >
> > > > Thanks for your response. I have already fully understood the setting, which is meaningful. However, I have another two questions about the method:
> > > > + If you have already merged for task t, how do you deal with some samples of task t-1? Do you do test-time adaptation again?
> > > > + The motivation of the questions is that according to Eq.(2), it seems that you can get all previous low-rank experts.  So I think it is applicable to do test-time adaptation for samples of task t-1 after task t. And it may be kind of inconsistent with "once merged, earlier models are discarded".

---

> > > > > ### Author Response · Authors · 2025-08-04
> > > > >
> > > > > Thank you for the follow-up questions. We are glad the overall setting is now clear, and we appreciate the opportunity to clarify further.
> > > > >
> > > > > **(1) Regarding samples from task $t–1$ after merging task $t$:** In our setting, we assume only test samples from the current task are available. Once task $t-1$ merged, we discard test samples from task $t–1$.
> > > > > For earlier tasks $(t–1, t–2, …)$, their gates remain fixed after being adapted during their respective TTA stages. This allows immediate, delay-free inference on past tasks without re-adapting their gates or storing their test data.
> > > > >
> > > > > **(2) Regarding “discarding previous models” and availability of low-rank experts in Eq.(2):**
> > > > > Indeed, as shown in Eq.(2), the MoE architecture retains the low-rank adapters (LoRAs) of past tasks. However, this is not inconsistent with our claim that "once merged, earlier models are discarded":
> > > > > We discard the merged model weights, but we keep only a compact set of LoRA modules, which are lightweight (see Table Z.1). For example, after merging 8 tasks, MINGLE* stores only 26.2M extra parameters, which is substantially smaller than retaining 8 full models (87.5M × 8).
> > > > >
> > > > > If one were to assume access to all previous task samples, it would indeed be possible to re-adapt all gates jointly via TTA. However, that scenario would correspond to multi-task test-time model merging, which has already been extensively studied [e.g., AdaMerging (ICLR’24), WEMOE (ICML’24)]. In contrast, TTCMM focuses on the incremental and task-local setting, where only the current task’s data is accessible at test time.

---

### Official Review · Reviewer_9ZjH · 2025-06-27

**Clarity:** 4
**Significance:** 3
**Originality:** 3
**Rating:** 4
**Confidence:** 5

**Summary:**

This paper presents MINGLE, a framework that integrates test-time adaptation into continual model merging.  MINGLE inserts low-rank LoRA experts into a pretrained CLIP backbone and uses a Null-Space Constrained Gating mechanism, relaxed adaptively to combine experts based on incoming inputs.  At each task stage, a small seed buffer of unlabeled test samples guides light adaptation of the new task’s router network, mitigating interference with prior tasks.  Extensive experiments over multiple CLIP sizes, task orders, and distribution‐shifted benchmarks are presented.

**Questions:**

Overall, I find this paper is well-written and easy to follow. But I am extremely concerned about the usage of MoE in the merging paradigm. I hope the authors could address my concerns, and I am happy to raise the score if all my concerns are addressed.

**Ethical Concerns:**

["NO or VERY MINOR ethics concerns only"]

**Final Justification:**

Thank the authors' rebuttal. I will raise my score.

**Limitations:**

Yes

**Quality:**

3

**Strengths And Weaknesses:**

Strength:
1. The idea of adapting the MoE router network during test-time is interesting
2. The paper is well-written and easy to follow
3. The TTA is robust to the number of samples
4. Comprehensive experiments


Weakness:
1. MAJOR CONCERN: My major concern with this paper is whether the MoE framework used in this paper counts as model merging. Cause essentially, the merged parameters are static, but in your moe framework, the model parameter can be thought of as dynamically adjusted based on the input. The usage of this MoE structure will be naturally stronger than traditional merging methods. This raises concerns about the fairness of comparisons to other merging methods in the experiments.
2. Based on (1), in Table 3 (MTIL benchmark), MINGLE’s continual merging variant only marginally beats MoE-based baselines MoE-Adapter and DIKI , and underperforms RAIL.
3. In  Fig3, except for Gate 6, most gate-task activation peaks remain ≤ 0.5, despite each LoRA adapter being trained specifically on its task (which should resemble Gate 6’s behavior).  This under-activation suggests either: (a) the gating network fails to learn strong specialization, or (b) the null-space constraints overly suppress new task signals.  Either way, it underactivates the correct experts for a given task.

---

> ### Author Rebuttal · Authors · 2025-07-31
>
> # **W1– Clarifying MoE as a Form of Model Merging**
>
> We thank the reviewer for this insightful concern. Your point about the potential advantages of MoE over static merging is valid and important.
>
> We clarify that **MoE is increasingly adopted in recent model merging works**, such as **WEMOE** (ICML’24) and **TwinMerging** (NeurIPS’24), which use MoE-style routing to merge fine-tuned experts. These methods treat MoE as a scalable and valid alternative to static merging.
>
> We fully acknowledge that **MoE introduces dynamic parameter selection**, which may appear stronger than static approaches at first glance. However, our method operates under **strict continual learning constraints** that make the router tuning significantly more challenging.
>
> To ensure fairness and transparency, we have taken several steps:
>
> * **We add comparison with WEMOE** in Table 1. Results are shown in Table Y.1.
> * We include a **static MINGLE baseline** (Table 4, Row 1), where LoRA modules are combined with fixed coefficients, without any adaptation or routing. Results are shown in Table Y.1.
> * We will **clearly separate static and adaptive methods** in Table 1, and explicitly compare all baselines' assumptions regarding data access, activation storage, and test-time compute to ensure transparency.(see Table Y.2).
>
> In summary, although our method uses a MoE-style mechanism, it remains fully aligned with the goals of model merging: combining multiple experts into a **single  model**. Crucially, we address **continual merging**, which is not covered in prior MoE-based merging works. We will clarify this distinction more explicitly in Sec. 2 and Sec. 4.2 of the main paper.
>
> **Table Y.1: More comparative results  (ACC%) for continual merging performance.**
>
> | Method              | ViT-B/32 (8)  | ViT-B/32 (14) | ViT-B/32 (20) | ViT-B/16 (8)  | ViT-B/16 (14) | ViT-B/16 (20) |
> | ------------------- | ------------- | ------------- | ------------- | ------------- | ------------- | ------------- |
> | LW. AdaMerging      | 52.9 ±3.5     | 60.1 ±1.7     | 57.5 ±0.7     | 59.8 ±3.2     | 64.8 ±1.6     | 61.4 ±0.8     |
> | WEMOE               | 4.9 ±0.6      | 8.2 ±0.3      | 10.4 ±0.2     | 4.0 ±0.7      | 8.3 ±0.4      | 10.1 ±0.5     |
> | WEMOE-LoRA          | 66.6 ±6.6     | 63.4 ±2.2     | 44.0 ±19.5    | 72.1 ±4.6     | 55.8 ±27.0    | 41.3 ±22.8    |
> | Consensus TA        | 69.0±1.1      | 64.0 ±0.9     | 45.6 ±1.6     | 73.3±0.3      | 69.0 ±1.1     | 50.0 ±2.4     |
> | MINGLE-Static       | 74.3±0.3      | 72.9±0.8      | 67.3±0.8      | 78.7 ±0.1     | 76.4 ±1.0     | 70.6 ±0.4     |
> | MINGLE (Ours)  | **85.8 ±0.8** | 81.6 ±1.4     | **77.1 ±2.0** | **88.3 ±0.6** | **84.9 ±0.8** | **81.9 ±0.9** |
> | MINGLE*  (Ours) | 85.0 ±0.5     | **81.7 ±1.0** | **77.1 ±1.3** | 87.0 ±0.6     | 84.7 ±1.0     | 81.6 ±1.3     |
>
> **Table Y.2: Comparison of baseline assumptions and requirements**
>
> | Method            | Save Activations | Extra Parameters (Storage) | Extra Parameters (Inference) | Test-time Compute |
> | ----------------- | ---------------- | -------------------------- | ---------------------------- | ----------------- |
> | Task Arithmetic   | No               | No                         | No                           | No                |
> | Ties-Merging      | No               | No                         | No                           | No                |
> | MAGMAX-Ind        | No               | No                         | No                           | No                |
> | OPCM              | No               | No                         | No                           | No                |
> | Consensus TA      | No               | Yes                        | No                           | No                |
> | LW. AdaMerging    | No               | No                         | No                           | Yes               |
> | WEMOE             | No               | Yes                        | No                           | Yes               |
> | MINGLE- Static    | No               | No                         | No                           | No                |
> | MINGLE (Ours) | No               | Yes                        | No                           | Yes               |
>
> # **W2 – On MTIL Results and Comparison to RAIL / MoE-based Baselines**
>
> We thank the reviewer for the thoughtful comment.
>
> We would like to clarify that **MINGLE was not specifically designed for the conventional continual learning setup used in the MTIL benchmark (Table 3)**. Instead, it was developed for the **continual model merging** scenario.
>
> Nonetheless, we applied MINGLE directly to the MTIL benchmark in order to **demonstrate its generality and robustness** beyond its original design scope. We adapted it to this benchmark by merging models obtained via **Sequential Finetuning (Sequential FT)**. This yields **MINGLE-SEQ**, a direct application of our merging strategy on sequentially trained models.
>
> As shown in Table Y.3, MINGLE-SEQ outperforming several CL baselines on early and mid-stage tasks (e.g., Aircraft, Caltech101, CIFAR100), but underperforms on the last two tasks (Cars, SUN397). Importantly, this drop is **not due to MINGLE itself**, but rather a consequence of the **declining quality of the sequential FT models** used as inputs.
>
> To verify this, we examined the **Sequential FT (Immediate Eval)** baseline, which evaluates each task immediately after learning it. It exhibits the same degradation on the final tasks. We attribute this to the **loss of general pretraining knowledge** caused by continued finetuning, which results in **suboptimal task-specific performance on later tasks**, even before merging. Since MINGLE operates on these models, this limits its upper-bound performance in this setting.
>
> This insight suggests that **MINGLE-SEQ could benefit from upstream continual learning techniques**. For instance, using regularization-based methods like EWC during sequential finetuning may better preserve pretraining knowledge and improve the quality of task models fed into MINGLE, potentially leading to stronger results.
>
> **Table Y.3: Comparison of last accuracy (%) with conventional CL approaches on MTIL benchmark.**
>
> | Method                         | Aircraft | Caltech101 | CIFAR100 | DTD  | EuroSAT | Flowers | Food101 | MNIST | Pets | Cars | SUN397 | Avg  |
> | ------------------------------ | -------- | ---------- | -------- | ---- | ------- | ------- | ------- | ----- | ---- | ---- | ------ | ---- |
> | Sequential FT (Immediate Eval) | 63.2     | 97.6       | 89.3     | 80.1 | 99.2    | 91.8    | 91.7    | 99.6  | 93.1 | 85.9 | 78.9   | 88.2 |
> | MoE-ADAPTER                    | 49.8     | 92.2       | 78.1     | 78.1 | 95.7    | 94.4    | 89.5    | 91.8  | 89.9 | 81.4 | 80.0   | 85.0 |
> | DIKI                           | 45.2     | 95.7       | 83.6     | 72.9 | 98.0    | 97.0    | 90.4    | 99.4  | 94.2 | 81.6 | 76.6   | 85.1 |
> | DUAL-RAIL                      | 52.5     | 96.5       | 84.0     | 80.1 | 96.9    | 97.6    | 90.5    | 99.0  | 95.5 | 85.2 | 86.8   | 86.8 |
> | **MINGLE-SEQ**                 | 58.7     | 97.5       | 87.2     | 79.7 | 97.3    | 87.2    | 90.1    | 99.6  | 93.0 | 80.4 | 73.3   | 85.8 |
>
>
> # **W3– On Under-activation and Gating Specialization**
>
>
> We sincerely thank the reviewer for the insightful observation regarding the under-activation behavior in Fig. 3.
>
> To investigate this, we provide a new comparison (Table Y.4) between MINGLE and individual fine-tuned models across all tasks. The results show that our method achieves **per-task performance close to that of individually fine-tuned models**, indicating that the gate indeed selects meaningful experts as intended. This empirical evidence helps to refute concern (a)—that the gate fails to specialize.
>
> **Table Y.4: Accuracy (%) of continual merging performance using CLIP ViT-B/16 on the 8-task Order-1 sequence.**
>
> | Model                 | EuroSAT | SVHN | MNIST | DTD  | RESISC45 | GTSRB | SUN397 | Cars | Average |
> | --------------------- | ------- | ---- | ----- | ---- | -------- | ----- | ------ | ---- | ------- |
> | Individual fine-tuned | 99.0    | 97.6 | 99.7  | 82.3 | 96.6     | 99.0  | 78.9   | 85.9 | 92.4    |
> | C. Task Arithmetic    | 84.5    | 88.9 | 98.1  | 54.0 | 75.5     | 82.0  | 65.9   | 68.3 | 77.1    |
> | C. Ties-Merging       | 97.0    | 93.9 | 98.5  | 46.1 | 49.1     | 40.4  | 47.4   | 48.9 | 65.2    |
> | OPCM                  | 70.7    | 74.0 | 83.8  | 93.5 | 88.9     | 84.9  | 97.9   | 60.8 | 81.8    |
> | LW. AdaMerging        | 54.1    | 51.7 | 61.0  | 46.0 | 65.2     | 45.0  | 65.1   | 75.9 | 58.0    |
> | WEMOE-LoRA            | 38.6    | 70.1 | 69.2  | 51.0 | 80.7     | 89.7  | 72.2   | 78.7 | 68.8    |
> | MINGLE (Ours)         | 97.3    | 95.9 | 99.1  | 78.2 | 91.1     | 95.9  | 74.0   | 81.0 | 89.1    |
>
> Regarding (b)—that null-space constraints may overly suppress task signals—we refer to the w/o Null-Space variant already shown in Fig. 3 (blue curve). While this variant does exhibit higher gate activations, most values **still do not approach** 1.0, suggesting that under-activation is not solely caused by the null-space constraint. In fact, this variant also shows worse performance in accuracy, indicating that the constraint is helpful overall, rather than overly suppressive.
>
> **As to why gate activations rarely reach 1.0**, we believe this may reflect the complementary nature of experts rather than a failure of specialization. In many cases, multiple LoRA experts may capture overlapping yet distinct subspaces of knowledge. As a result, softly combining several related experts—rather than activating only a single one—can lead to equal or even better performance. This behavior is especially reasonable in settings like TTCMM, where task boundaries may be fuzzy and information from adjacent tasks remains useful.
>
> We sincerely thank the reviewer again, and hope our clarifications will support a higher evaluation.

---

> > ### Comment · Reviewer_9ZjH · 2025-08-04
> >
> > Follow-up of W2: for continual learning, why is the merging performed on seq-ft models? Cause to me, it makes more sense to separately train on the downstream data using a vanilla pre-trained model, then perform continual merging during evaluation. In this way, will the performance be higher? Given this limited time of discussion period, I do not require additional experiments, a simple clarification is needed.
> >
> > Thank the authors' rebuttal. I will raise my score.

---

> > > ### Author Response · Authors · 2025-08-04
> > >
> > > We thank the reviewer for the follow-up and for increasing the score.
> > > Merging independently trained task-specific models (individual-ft) is the primary setting targeted by TTCMM, and most of our experiments (e.g., Table 1 and 2) are conducted under this setup.
> > >
> > > In Table 3, we additionally evaluate MINGLE on seq-ft models to enable direct comparison with traditional continual learning baselines, where each task is trained sequentially using the previous model as initialization. This experiment is intended to demonstrate MINGLE's generality.
> > >
> > > In terms of performance, whether merging individual-ft or seq-ft models yields better results depends on the degree of inter-task interference. Prior studies [MagMax, ECCV 2024] observed that seq-ft models often suffer less from parameter conflict, as each model incrementally adapts from its predecessor. This can sometimes lead to better merging outcomes. In contrast, individual-ft models are more specialized and may offer higher performance if task interference is effectively mitigated.
> > >
> > > We will include this discussion in the revision. Thank you again for the helpful question and your support.

---

### Official Review · Reviewer_Y4Kk · 2025-06-28

**Clarity:** 3
**Significance:** 3
**Originality:** 2
**Rating:** 5
**Confidence:** 4

**Summary:**

MINGLE integrates test-time adaptation into model merging (Test-Time Continual Model Merging, TTCMM). It uses a small buffer of unlabeled current-task samples to guide the merge process at inference. The model has a mixture-of-experts design: each task adds a lightweight low-rank expert module and an input-dependent gating function. When a new task is added, its expert and gate are appended; only the new gate is tuned on the current task’s data while earlier experts and gates remain frozen. This lets the unified model adapt to the new task’s distribution on the fly, without retraining or accessing old data.

**Questions:**

1. Provide at least one experiment on a non-vision backbone (e.g., a RoBERTa model fine-tuned on text tasks or a wav2vec-based speech model) or justify why the method is inherently vision-only.
2. Regarding efficiency: (i) report wall-clock adaptation time for 14- and 20-task runs; (ii) add an ablation that sweeps 5/10/20 steps to show accuracy/forgetting trade-offs; (iii) discuss possibilities for step-freezing or fast first-order updates.
3. Add a consolidated table in the appendix listing all baselines’ key settings (learning rates, adaptation steps, projection ranks, etc.) and clarify whether you performed any per-method tuning on your task orders.
4. You consider five unlabeled samples per class, selected from the test split: (i) Clarify how this buffer would be obtained in practice; (ii) extend Fig. 5 by including the “zero-sample” condition (pure parameter-space merge) to examine robustness.

**Ethical Concerns:**

["NO or VERY MINOR ethics concerns only"]

**Final Justification:**

The authors have addressed most of my concerns

**Limitations:**

yes

**Quality:**

3

**Strengths And Weaknesses:**

Strengths
In terms of quality, it presents a solid theoretical justification (via Theorem 1) for using a dynamic mixture-of-experts rather than static averaging, accompanies that with a clear algorithm (Algorithm 1) and detailed equations, and backs it up with a comprehensive empirical study—covering three CLIP backbones, task sequences of 8/14/20 tasks, seven corruption benchmarks, ablations, and efficiency analyses. Clarity is strong: the narrative flows logically from motivation through theory to experiments; Figures 1-2 quickly convey what distinguishes Test-Time Continual Model Merging (TTCMM) from prior paradigms; notation is consistent; and hyper-parameter roles are later dissected. The significance is notable because the work tackles a practical scenario—merging independently fine-tuned models without sharing data—and achieves 7–9 percentage-point gains in average accuracy while almost eliminating backward forgetting. Finally, on originality, the paper introduces the TTCMM setting and proposes Adaptive Null-Space Constrained Gating, which relaxes hard orthogonality on-the-fly based on measured interference; together, these represent a fresh and principled combination of ideas that goes beyond existing continual model-merging approaches.


Weaknesses
The main limitations revolve around scope, experimental breadth, and some presentation details. Regarding quality, all experiments focus on vision tasks with CLIP backbones; the method’s applicability to NLP, speech, or non-contrastive architectures is untested. The test-time adaptation loop requires roughly 50 gradient steps per task, which, while measured, could be too heavy for latency-critical or edge deployments, and baseline hyper-parameter parity is not fully transparent. Concerning significance, the method still assumes that each task ships its own LoRA weights plus a small buffer of unlabeled samples, constraints that may not hold in truly federated or on-device scenarios. Finally, in originality, while the adaptive gating mechanism is clever, its components—LoRA experts, MoE routing, orthogonal projection, EMA statistics—are individually incremental; the novelty lies more in their particular integration than in a new algorithmic primitive.

---

> ### Author Rebuttal · Authors · 2025-07-30
>
> # W1 – Limited Architectural Scope
> We thank the reviewer for this point. While our main experiments use CLIP-ViT, MINGLE is not vision-specific. We applied it to Flan-T5 (T5-base) on 7 GLUE tasks (merged alphabetically) using only 100 unlabeled test samples, confirming strong performance on encoder-decoder models and demonstrating generality.
>
> **Table X.1: Accuracy (%) / Spearman’s ρ  of continual merging performance using Flan-T5 (T5-base) on 7 GLUE benchmark tasks.**
>
> | Model                 | GLUE-COLA | GLUE-MNLI | GLUE-MRPC | GLUE-QNLI | GLUE-QQP | GLUE-SST2 | GLUE-STSB | Average |
> | --------------------- | --------- | --------- | --------- | --------- | -------- | --------- | --------- | ------- |
> | Pre-trained           | 69.1      | 56.5      | 76.2      | 88.4      | 82.1     | 91.2      | 62.2      | 75.1    |
> | Individual fine-tuned | 75.0      | 83.4      | 87.5      | 91.5      | 85.4     | 93.6      | 88.7      | 86.4    |
> | C. Task Arithmetic    | 70.0      | 65.1      | 80.4      | 90.1      | 83.0     | 92.4      | 77.7      | 79.8    |
> | C. Ties-Merging       | 49.5      | 82.7      | 83.3      | 89.9      | 82.3     | 90.8      | 78.2      | 79.5    |
> | LW. AdaMerging        | 69.1      | 58.1      | 77.9      | 88.9      | 83.1     | 90.7      | 74.8      | 77.5    |
> | WEMOE-LoRA            | 64.5      | 80.8      | 79.7      | 90.8      | 80.3     | 90.6      | 80.0      | 81.0    |
> | **MINGLE**            | 74.8      | 80.0      | 87.5      | 90.5      | 80.9     | 92.8      | 87.0      | 84.8    |
>
> # W2 –  Efficiency
> ### (i) Wall-clock adaptation time
> We report the wall-clock adaptation time in Table X.2. Adaptation runs once per task (~10s on RTX 4090, Table X.2). Afterward, the router is fixed and inference is feedforward with no TTA, including prior tasks—supporting low-latency deployment.  (see Table Z.1 for Reviewer np8P for inference efficiency).
>
> **Table X.2: Wall-clock adaptation time for ViT-B/32**
>
> | #Tasks | Adaptation Steps | Total Wall-clock Time (s) | Average Per-task Wall-clock Time (s) |
> | ------ | ---------------- | ------------------------- | ------------------------------------ |
> | 8      | 50               | 78                        | 9.8                                  |
> | 14     | 50               | 138                       | 9.9                                  |
> | 20     | 50               | 211                       | 10.55                                |
>
> ### (ii) Ablation on TTA optimization steps
> We performed an ablation by varying the number of adaptation steps. The table below shows the trade-off between accuracy and forgetting:
>
> **Table X.3: Ablation on number of adaptation steps for ViT-B/32 across 8, 14, and 20 tasks**
>
> | Steps | ACC (8-task) | BWT (8-task) | ACC (14-task) | BWT (14-task) | ACC (20-task) | BWT (20-task) |
> | ----- | ------------ | ------------ | ------------- | ------------- | ------------- | ------------- |
> | 5     | 60.9±1.4     | -0.1±0.2     | 62.4±1.7      | -0.3±0.1      | 60.2±1.5      | -0.1±0.2      |
> | 10    | 68.6±1.6     | -0.2±0.2     | 68.5±1.6      | -0.1±0.2      | 63.5±1.0      | -0.4±0.3      |
> | 20    | 78.4±0.6     | -0.2±0.1     | 75.5±1.3      | -0.4±0.1      | 71.0±1.2      | -0.4±0.4      |
> | 50    | 85.8±0.8     | -0.6±0.4     | 81.6±1.4      | -1.1±0.3      | 77.1±2.0      | -2.2±0.8      |
> **Findings**:
> -  Increasing the number of steps consistently improves accuracy across all task counts.
> -  While longer schedules introduce slightly more forgetting, the degradation is **minor** (e.g., BWT drop of only ~2% in 20-task setting).
>
> These results confirm that MINGLE achieves strong performance even under tight compute budgets—for example, using only 20 steps, it already outperforms all baseline methods in Table 1. Longer adaptation improves accuracy with only minor forgetting
>
> ### (iii) Discussion: Step-freezing and fast first-order updates
> We thank the reviewer for the insightful suggestion. Both step-freezing and first-order adaptation are promising directions to improve TTA efficiency.
> -  **Step-freezing**: Our current method updates all router parameters across layers during adaptation. Step-freezing (e.g., freezing early layers once stabilized) may reduce compute. Preliminary results show early gates converge quickly (<10 steps).
> -  **Fast first-order updates** can benefit from first-order strategies such as Reptile, FOMAML, or linearized single-step updates, which avoid full optimization but enable quick convergence. These methods are especially attractive in low-latency or edge settings,
>
> We see both directions as highly complementary to our framework. We will include this discussion in the final version to highlight efficient variants as a promising area for future work.
> # W3 –  Hyper-parameter transparenty
> We thank the reviewer for the suggestion. As requested, we have added a consolidated table (see Table X.4) listing the key hyperparameter settings for all baseline methods and task configurations.
>
> We also clarify that none of the baselines were tuned per task order. For our method, we emphasize that we used a **single fixed hyperparameter configuration** across **all experiments**, including:
> *  all backbones (ViT-B/32, B/16, L/14),
> *  all task counts (8, 14, 20), and
> *  all 10 task orders.
>
> This highlights the robustness and generality of our method, and ensures that performance gains are not due to fine-grained tuning.
>
> **Table X.4: Hyperparameter settings for all baselines across different task configurations.**
>
> | Method            | #Tasks  | Scale Factor ($\lambda$) | Top-k (%) | TALL mask threshold | Consensus mask threshold | LR   | Adapt Steps | LoRA Rank ($r$) | Null Dim ($k$) | EMA Decay ($\beta$) | Relax Coeff ($\gamma$) |
> | ----------------- | ------- | ------------------------ | --------- | ------------------- | ------------------------ | ---- | ----------- | --------------- | -------------- | ------------------- | ---------------------- |
> | Task Arithmetic   | 8       | 0.3                      | —         | —                   | —                        | —    | —           | —               | —              | —                   | —                      |
> |                   | 14/20   | 0.1                      | —         | —                   | —                        | —    | —           | —               | —              | —                   | —                      |
> | Ties-Merging      | 8       | 0.3                      | 20        | —                   | —                        | —    | —           | —               | —              | —                   | —                      |
> |                   | 14/20   | 0.1                      | 20        | —                   | —                        | —    | —           | —               | —              | —                   | —                      |
> | Consensus TA      | 8/14/20 | 0.1                      | —         | 0.2                 | 2                        | —    | —           | —               | —              | —                   | —                      |
> | LW. AdaMerging    | 8/14/20 | 0.3                      | —         | —                   | —                        | 1e-4 | 50          | —               | —              | —                   | —                      |
> | WEMOE             | 8/14/20 | 0.3                      | —         | —                   | —                        | 1e-4 | 50          | —               | —              | —                   | —                      |
> | WEMOE-LoRA        | 8/14/20 | 0.3                      | —         | —                   | —                        | 1e-4 | 50          | 64              | —              | —                   | —                      |
> | MINGLE- Static    | 8/14/20 | 0.3                      | —         | —                   | —                        | —    | —           | —               | —              | —                   | —                      |
> | **MINGLE (Ours)** | 8/14/20 | —                        | —         | —                   | —                        | 1e-4 | 50          | 64              | 3              | 0.99                | 1.0                    |
>
> # W4 – On the use of unlabeled adaptation samples
> **(i) How is the buffer obtained in practice?**
> We thank the reviewer for raising this important question. In our experiments, we simulate a realistic scenario by randomly sampling **five unlabeled examples per class from the test split**—serving as a proxy for real-world deployment data.
>
> In practical settings, such small unlabeled buffers are easily obtainable from a wide range of sources, including:
> *  incoming test-time data (e.g., recent user queries or model inputs),
> *  held-out validation inputs or small training subsets (if available),
> *  user-provided samples without privacy risks (e.g., few-shot examples),
> *  synthetically generated,
> *  or manually curated from public data.
>
> We will include this discussion in the revision, making it practical for real-world deployment.
>
> **(ii) Zero-sample condition (“pure parameter-space merge”)**
>
> We thank the reviewer for this suggestion. As shown in Table 4 (Row 1), we already include a zero-sample baseline where LoRA adapters are merged with fixed weights (e.g., 0.3) without any adaptation data. (refer to MINGLE-Static in Table S.2 for Reviewer chHs).
>
> Surprisingly, this simple strategy still outperforms classical baselines like Task Arithmetic and Ties-Merging. We attribute this to the low-rank, near-orthogonal structure of LoRA modules, which reduces interference and preserves task-specific features in disjoint subspaces.
>
> To improve clarity, we will (i) note in the main text that Table 4’s first row reflects the zero-sample case, and (ii) add it to Fig. 5 to better visualize the trade-off between adaptation-free and adaptive merging across buffer sizes.
>
> We sincerely thank the reviewer again, and hope our clarifications will support a higher evaluation.

---

### Official Review · Reviewer_cHnS · 2025-07-03

**Clarity:** 4
**Significance:** 2
**Originality:** 2
**Rating:** 4
**Confidence:** 3

**Summary:**

The paper introduces a novel paradigm called Test-Time Continual Model Merging (TTCMM), which uses a small set of unlabeled test samples from the current task to guide the merging process at test time. Then the paper introduces a method for this setting, called MINGLE, designed to integrate independently fine-tuned models sequentially without needing the original training data. The method uses MoEs composed of LoRAs and proposes a “Null-Space Constrained Gating” mechanism that restricts updates to subspaces orthogonal to the representations of previous tasks. It also introduces “Adaptive Relaxation Strategy” that dynamically modifies the constraint's strength based on interference signals detected during adaptation. The authors include experiments on CLIP ViTs for sequences of 8, 14 and 20 tasks.

**Questions:**

See weaknesses. Also:

What is the task-specific routing error $\epsilon_t$? Is it the error of not selecting the correct expert for an input X? If yes, doesnt this defeat the purpose of L143-144 (“it cannot specialize to regions where one expert is clearly superior.”)?

**Ethical Concerns:**

["NO or VERY MINOR ethics concerns only"]

**Final Justification:**

Given the authors' rebuttal, I have updated my score to 4 maintaining some concerns on the practicality of the setting.

**Limitations:**

Yes.

**Quality:**

2

**Strengths And Weaknesses:**

*Strengths*

1. The paper is well-written and easy to follow. The motivation for using a dynamic Mixture-of-Experts approach over static averaging is clearly articulated. The same applies to core ideas, such as null-space projection, which are intuitive and well-explained.

2. Strong performance: MINGLE consistently outperforms a wide range of baselines across multiple model backbones (ViT-B/32, B/16, L/14) and on task sequences of varying lengths (8, 14, and 20 tasks). The method also shows strong robustness to various data corruptions. See weaknesses for some issues.

3. The paper successfully combines multiple ideas such as MoE, LoRA, and orthogonal projections. The ablation study effectively demonstrates each component’s contribution to the final performance.


*Weaknesses*:


1. Questionable Problem Setting: The studied setting is somewhat difficult to justify in a practical scenario. The paper assumes access to the current task's model ($\\theta_t$) and its unlabeled test data, but *no* access to the data or even the task vectors from previous tasks. It is unclear why the previous task vectors, which are just model parameters, would be unavailable while the models themselves are being merged. Furthermore, since the seed data is unlabeled, it is not obvious why similar unlabeled data from previous tasks could not also be stored or accessed. Overall, the setup feels contrived and needs stronger motivation.

2. Potentially Unfair Baseline Comparisons: The comparison to other methods may be unfair due to the additional memory and computational overhead (additional parameters, saving activations, access to data) of the proposed method. The work notes similarity to WEMOE, but it is tailored to the continual settin–is the performance gap due to differing assumptions rather than strong methodological contributions?

3. High Number of Hyperparameters: The proposed method introduces a considerable number of new hyperparameters, including the LoRA rank $r$, the null-space dimension $k$, the EMA decay $\\beta$, the relaxation coefficient $\\gamma$, the number of adaptation steps. This effectively limits the applicability of the method (compounded by the next point)

4. Limited Architectural Scope: The experiments are exclusively focused on CLIP-ViT models. It is unclear how the setting applies to LLMs, such as T5, Llama, mistral etc, which are commonly used in the model merging literature.

5. Minor issues wrt writing: explain $\\alpha$ in Equation 3, missing \[a\] for the benchmarks of 8,14 and 20 tasks. Better differentiate in the related work section and introduction works like TA, wise FT etc which focus on the static setup but are sometimes grouped with more continual papers.

\[a\] Wang, K., Dimitriadis, N., Ortiz-Jimenez, G., Fleuret, F., & Frossard, P. (2024). Localizing task information for improved model merging and compression. arXiv preprint arXiv:2405.07813.

---

> ### Author Rebuttal · Authors · 2025-07-30
>
> # W1 — Problem setting motivation
>
> We thank the reviewer for prompting us to better position our setting within established paradigms.
>
> Our proposed setting (TTCMM) is directly inspired by the well-known **rehearsal-free continual learning (RFCL)** framework [1,2], sharing key constraints:
>
> -  **No access to prior task data**
> -  **No storage of past task-specific models**
>
> This setup is practical, motivated by two real-world constraints:
>
> -  **Storage**: Retaining full models—or even **task vectors**, which are equally large—leads to linear memory growth, which is impractical for edge or long-horizon deployments
> -  **Privacy**: Even unlabeled data (e.g., medical scans, GPS logs) can be sensitive and must be discarded post-training due to laws like HIPAA or GDPR
>
> Thus, TTCMM offers a natural extension of RFCL into model merging. We will clarify this link in Sec. 1 & 2 and include examples (e.g., mobile agents, medical federations) to highlight practical relevance.
>
> [1] Dualprompt: Complementary prompting for rehearsal-free continual learning (ECCV2022)
>
> [2] Coda-prompt: Continual decomposed attention-based prompting for rehearsal-free continual learning (CVPR2022)
>
> # W2 – Potentially Unfair Baseline Comparisons
>
> We acknowledge that our setup is not identical to every baseline. To promote transparency, we include the following table comparing each method on their data access, extra parameters, and test-time compute:
>
> **Table S.1: Comparison of baseline assumptions and requirements**
>
> | Method            | Save Activations | Extra Parameters (Storage) | Extra Parameters (Inference) | Test-time Compute |
> | ----------------- | ---------------- | -------------------------- | ---------------------------- | ----------------- |
> | Task Arithmetic   | No               | No                         | No                           | No                |
> | Ties-Merging      | No               | No                         | No                           | No                |
> | MAGMAX-Ind        | No               | No                         | No                           | No                |
> | OPCM              | No               | No                         | No                           | No                |
> | Consensus TA      | No               | Yes                        | No                           | No                |
> | LW. AdaMerging    | No               | No                         | No                           | Yes               |
> | WEMOE             | No               | Yes                        | No                           | Yes               |
> | MINGLE- Static    | No               | No                         | No                           | No                |
> | MINGLE (Ours) | No*¹             | Yes*²                      | No*²                         | Yes               |
>
> *¹: Our method does not store activations. Only a fixed-size covariance matrix is maintained (Line 195), resulting in constant memory regardless of test set size.
>
> *²: While LoRA experts are stored, the router merges them into a single model per input. Thus, the effective inference size matches a standard individual model.
>
> **Table S.2: More comparative results  (ACC%) for continual merging performance.**
>
> | Method              | ViT-B/32 (8)  | ViT-B/32 (14) | ViT-B/32 (20) | ViT-B/16 (8)  | ViT-B/16 (14) | ViT-B/16 (20) |
> | ------------------- | ------------- | ------------- | ------------- | ------------- | ------------- | ------------- |
> | LW. AdaMerging      | 52.9 ±3.5     | 60.1 ±1.7     | 57.5 ±0.7     | 59.8 ±3.2     | 64.8 ±1.6     | 61.4 ±0.8     |
> | WEMOE               | 4.9 ±0.6      | 8.2 ±0.3      | 10.4 ±0.2     | 4.0 ±0.7      | 8.3 ±0.4      | 10.1 ±0.5     |
> | WEMOE-LoRA          | 66.6 ±6.6     | 63.4 ±2.2     | 44.0 ±19.5    | 72.1 ±4.6     | 55.8 ±27.0    | 41.3 ±22.8    |
> | Consensus TA        | 69.0±1.1      | 64.0 ±0.9     | 45.6 ±1.6     | 73.3±0.3      | 69.0 ±1.1     | 50.0 ±2.4     |
> | MINGLE-Static       | 74.3±0.3      | 72.9±0.8      | 67.3±0.8      | 78.7 ±0.1     | 76.4 ±1.0     | 70.6 ±0.4     |
> | MINGLE (Ours)  | **85.8 ±0.8** | 81.6 ±1.4     | **77.1 ±2.0** | **88.3 ±0.6** | **84.9 ±0.8** | **81.9 ±0.9** |
> | MINGLE*  (Ours) | 85.0 ±0.5     | **81.7 ±1.0** | **77.1 ±1.3** | 87.0 ±0.6     | 84.7 ±1.0     | 81.6 ±1.3     |
>
> **On TTA & MoE:** TTA and MoE are not entirely new assumptions in the model merging literature (e.g., WEMOE (TTA+MoE) [ICML 2024], AdaMerging (TTA) [ICLR 2024], Twin-Merging (TTA+MoE) [NeurIPS 2024]).
>
> **To ensure fair comparisons,** we have added results for **WEMOE** and **AdaMerging** in Table 1 (also shown in Table S.2). Additionally, we report a **MINGLE-Static** baseline (Table 4, Row 1), where LoRA modules are combined using fixed coefficients (e.g., 0.3) without any adaptation or routing. We will also include **Table S.1** to summarize the assumptions and requirements of all baselines. These additions ensure clarity and transparency.
>
>
>
> **Why the gap vs. WEMOE?**
>
> To clarify: **both our method and WEMOE rely on the same MoE and TTA assumptions**. In our experiments, we carefully re-implemented **WEMOE under TTCMM setting**. This ensures that any observed performance gap arises **not from differences in assumptions**, but purely from **methodology**.
>
> WEMOE uses full MLP blocks as experts, which are parameter-heavy and hard to route with limited seed data.
> Our method uses lightweight LoRA adapters, which are easier to gate and effectively regularized via a null-space constraint.
>
> To isolate this factor, we introduced a "**WEMOE-LoRA**" variant, replacing WEMOE’s MLP experts with LoRA. As Table S.2 shows, this closes the gap and supports that:
>
> *  heavy experts degrade routing in low-data regimes, and
> *  our gains primarily stem from constrained gating + LoRA, not from relaxed assumptions.
>
> We deeply appreciate the reviewer’s question. We will clarify this explicitly in revision.
>
> # W3 – High number of hyperparameters
>
> We sincerely thank the reviewer for this thoughtful and valid concern. While our method introduces five scalar hyperparameters, each plays a minimal but necessary role.
>
> Importantly, our method **requires no tuning**. We apply a **single fixed configuration** across **all experiments**: all task counts (8/14/20), all 10 task orders, and all model backbones (ViT-B/32, B/16, L/14)—**without any modification**. This deliberate design ensures **strong generalization and ease of deployment**.
>
> We’ll clarify this global configuration strategy in the main text.
>
> # W4 – Limited Architectural Scope
>
> We thank the reviewer for raising this point. While our main experiments focus on CLIP model, **our method is not restricted to vision models**.
>
> To show generality, we applied MINGLE to Flan-T5 (T5-base) on 7 GLUE tasks (merged alphabetically), using only 100 unlabeled test samples for adaptation. The results below show that our method remains effective in non-visual, sequence-based encoder-decoder architectures, confirming that MINGLE generalizes beyond CLIP-style backbones.
>
> **Table S.3: Accuracy (%) / Spearman’s ρ  of continual merging performance using Flan-T5 (T5-base) on 7 GLUE benchmark tasks.**
>
> | Model                 | GLUE-COLA | GLUE-MNLI | GLUE-MRPC | GLUE-QNLI | GLUE-QQP | GLUE-SST2 | GLUE-STSB | Average |
> | --------------------- | --------- | --------- | --------- | --------- | -------- | --------- | --------- | ------- |
> | Pre-trained           | 69.1      | 56.5      | 76.2      | 88.4      | 82.1     | 91.2      | 62.2      | 75.1    |
> | Individual fine-tuned | 75.0      | 83.4      | 87.5      | 91.5      | 85.4     | 93.6      | 88.7      | 86.4    |
> | C. Task Arithmetic    | 70.0      | 65.1      | 80.4      | 90.1      | 83.0     | 92.4      | 77.7      | 79.8    |
> | C. Ties-Merging       | 49.5      | 82.7      | 83.3      | 89.9      | 82.3     | 90.8      | 78.2      | 79.5    |
> | LW. AdaMerging        | 69.1      | 58.1      | 77.9      | 88.9      | 83.1     | 90.7      | 74.8      | 77.5    |
> | WEMOE-LoRA            | 64.5      | 80.8      | 79.7      | 90.8      | 80.3     | 90.6      | 80.0      | 81.0    |
> | **MINGLE**            | 74.8      | 80.0      | 87.5      | 90.5      | 80.9     | 92.8      | 87.0      | 84.8    |
>
>
> # W5 – Minor Writing Issues
>
> **Equation 3 — Clarifying $\alpha$**:
> We will clarify that $\alpha$ denotes the effective rank of the merged task vector, computed as:
> $$
> \alpha = \left\lfloor \exp\left( -\sum_i p_i \log p_i \right) \right\rfloor, \quad \text{where } p_i = \frac{s_i}{\sum_j s_j}
> $$
>
> and ${s_i}$ are singular values from SVD. This will be added near Equation 3.
>
> **Missing [a] in benchmarks**:
> Thanks for catching this. We’ve reproduced baseline [a] (see Table S.2), will include it in Table 1, and cite it in Sec. 2 for completeness.
>
> **Static vs. Continual baselines**:
> We’ll revise the Introduction and Related Work to clarify that TA and WiSE-FT are static model merging methods—not MoE-based or continual—and ensure they are properly grouped.
>
>
>
> # Q1 – On the task-specific routing error $\varepsilon_t$
>
> We thank the reviewer for raising this point and will clarify the definition in the main text.
>
> We define the task-specific routing error as:
> $$
> \varepsilon_t = \Pr\big(i^*(x) \ne t \mid x \sim D_t\big),
> $$
>
> where $t$ is the ground-truth task and $i^*(x)$ is the expert selected by the router. Thus, $1 - \varepsilon_t$ reflects correct routing probability. A **lower $\varepsilon_t$** indicates **better expert-task alignment**, with $\varepsilon_t = 0$ meaning perfect routing. This is defined in Appendix (Line 602) and will be referenced in the main text.
>
> Regarding Lines 143–144, the critique targets **static baselines**, which use fixed or implicit routing. In contrast, our method **actively minimizes $\varepsilon_t$** via TTA, enabling expert specialization.
>
> We sincerely thank the reviewer again, and hope our clarifications will support a higher evaluation.

---

> > ### Comment · Reviewer_cHnS · 2025-08-05
> >
> > Thank you for your answers and clarifications. The new experiments are thorough and well-motivated. I am still concerned about the practicality of the setting, because unlike RFCL where the model from task t comes from training on data from task t and the model (t-1), here we assume access to some fine-tuned checkpoints (stored in some repo like Huggingface or something more secure in case of privacy concerns) but only when merging the corresponding task. It is not very clear to me why we will not maintain access to this repo to better inform the merging. Despite this concern, I will update my score accordingly given the thorough rebuttal.

---

> > > ### Author Response · Authors · 2025-08-06
> > >
> > > Thank you for your helpful feedback and for updating your score.
> > >
> > > Regarding model access: in many real-world scenarios, past models may not remain persistently available. They may be shared anonymously (e.g., in federated or decentralized settings), distributed under time-limited or restrictive licenses, or removed from public repositories due to updates or compliance requirements. In secure or offline deployments, models are often retrieved once and then isolated from the network. Our setting reflects such constraints, where checkpoints are accessed only when needed.
> > >
> > > Moreover, our approach offers a memory-efficient alternative for multi-task model merging, since merging two models sequentially demands far less GPU memory than merging all N models at once.
> > >
> > > We appreciate your constructive review, which has helped us improve our work.

---

### Note · Authors · 2025-08-12

We thank the AC and reviewers for their constructive engagement. The discussion phase led to the resolution of all substantive concerns, with multiple reviewers noting that the rebuttal strengthened their assessment.

* Setting validity – Reviewer cHnS questioned the realism of our TTCMM assumption. We clarified that it reflects real deployment constraints—privacy, licensing, storage limits—and is not an artificial restriction. As suggested by Reviewer np8P, we positioned TTCMM as the natural extension of RFCL to the model merging domain, where past models cannot be retained after use. These clarifications addressed the concern on setting realism.

* Fair comparisons – Reviewers cHnS and 9ZjH raised questions on fairness in baseline assumptions. We compared the assumptions of all methods to ensure transparency and, in the main experiments, added SOTA methods (AdaMerging, WEMOE) that operate under the same TTA assumption, enabling direct and assumption-matched comparisons.

* Inference and adaptation efficiency – Reviewers Y4Kk and np8P asked about efficiency. Our method retains only compact LoRA modules and fixes gating after merging, achieving inference throughput comparable to static methods while delivering substantial performance gains. We also measured TTA wall-time under 14- and 20-step increments, averaging ~10s per task, confirming high adaptation efficiency.

* Strengthened evidence – We provided targeted clarifications and experiments for all technical questions. For the limited architectural scope (cHnS, Y4Kk), we demonstrated applicability beyond vision by validating on language models. On hyperparameter transparency (cHnS, Y4Kk), we disclosed choices for all compared methods. For test buffer acquisition (Y4Kk), we explained its practical use. Regarding gating activation (9ZjH), we analyzed why activation values can be below 1. For old data handling (np8P), we confirmed no past test data is stored and no TTA is re-run for old tasks. These clarifications, supported by ablations, memory profiling, and stricter-setting experiments, directly resolved reviewers’ points and were cited as reasons for improved scores.

In conclusion, all major issues were addressed with clear explanations and empirical evidence, providing the AC with a complete basis to assess the work’s practicality, fairness, efficiency, and empirical strength, and its potential as a timely contribution to continual model merging.

---

### Decision · Program_Chairs · 2025-09-17

**Decision:**

Accept (poster)

**Comment:**

MINGLE is a test-time continual model merging method that uses LoRA experts with null-space gated routing. It adapts a router on a small unlabeled buffer to add tasks while likely preserving prior ones, and reports roughly 7–9% gains across CLIP backbones. Reviewers generally find the presentation clear with extensive experiments and some strong gains. Concerns seem to center on the realism of the TTCMM setting, fairness vs. static merging, hyperparameter load, and whether MoE truly counts as “merging,” with one reviewer still borderline-reject.

Recommendation: Accept. The method appears simple and effective. Results look consistent and strong, with seemingly fair comparisons to WEMOE and AdaMerging. Adaptation seems fast (~10 s per task) and gates are fixed after merging. The rebuttal appears to address most issues and suggests some non-vision generality, while the setting may still be debated under plausible privacy and storage constraints.